# Matting Anything 2: Towards Video Matting for Anything

**Chenyi Zhang[1], Yiheng Lin[2], Yunchao Wei[2], Hongsong Wang[3], Caifeng Shan[1], Fang Zhao[1]** [*]
[1]Nanjing University, [2]Beijing Jiaotong University, [3]Southeast University
zchenyi007@gmail.com, 23120298@bjtu.edu.cn, wychao1987@gmail.com
hongsongwang@seu.edu.cn, cfshan@nju.edu.cn, fzhao@nju.edu.cn

## Abstract

Video matting is a crucial task for many applications, but existing methods face significant limitations. They are often domain-specific, focusing primarily on human portraits, and rely on the mask of first frame that is challenging to acquire for transparent or intricate objects like fire or smoke. To address these challenges, we introduce Matting Anything 2 (MAM2), a versatile and robust video matting model that handles diverse objects using flexible user prompts such as points, boxes, or masks. We first propose Promptable Dual-mode Decoder (PDD), an effective structure that simultaneously predicts a segmentation mask and a corresponding high-quality trimap, leveraging trimap-based guidance to improve generalization. To tackle prediction instability for transparent objects across video frames, we further propose a Memory-Separable Siamese (MSS) mechanism. MSS employs a recurrent approach that isolates trimap prediction from potentially interfering mask memory, significantly enhancing temporal consistency. To validate our method's performance on diverse objects, we introduce the Natural Object Video Matting dataset, a new benchmark with substantially greater diversity. Extensive experiments show that MAM2 possesses exceptional matting accuracy and generalization capabilities. We believe MAM2 demonstrates a significant leap forward in creating a video matting method for anything. The code is available at Matting-Anything-2.

## 1 Introduction

Video matting, the process of precisely extracting the foreground alpha matte from a video sequence, is a critical enabling technology for a myriad of applications. It is fundamental to the visual effects industry for seamless cinematic composition, powers virtual backgrounds in video conferencing, and facilitates realistic object integration in augmented reality experiences.

Despite significant advancements in the field, existing video matting methods still exhibit several critical limitations: i) Domain Specificity: The vast majority of recent video matting models are predominantly human-centric Lin et al. (2021b); Huynh et al. (2024); Yang et al. (2025); Li et al. (2024a); Ke et al. (2022); Ge et al. (2025), focusing almost exclusively on human portrait. Research into matting for more general natural scenes remains largely underexplored, a stark contrast to the well-developed state of natural image matting. Concurrently, as illustrated in Table 1, a comprehensive benchmark for evaluating the generalization capabilities of video matting models on diverse natural scenes is conspicuously absent. ii) Reliance on Mask: Popular methods often follow the standard semi-supervised Video Object Segmentation (VOS) framework Yang et al. (2021); Caelles et al. (2017); Yang et al. (2025). This framework necessitates user interaction on the first frame, typically by providing a mask to specify the target object. However, a high-quality mask cannot always be easily obtained via an interactive segmentation model, particularly for certain types of objects. For transparent objects (e.g., smoke, fire), many regions exhibit a high degree of transparency that is difficult for both the human eye and the model to discern, making such a mask challenging to acquire. In this case, a bounding box is clearly a more efficient form of interaction, as the user only needs to roughly enclose the object, eliminating the need for pixel-level mask correction.

---

[*]Corresponding author.

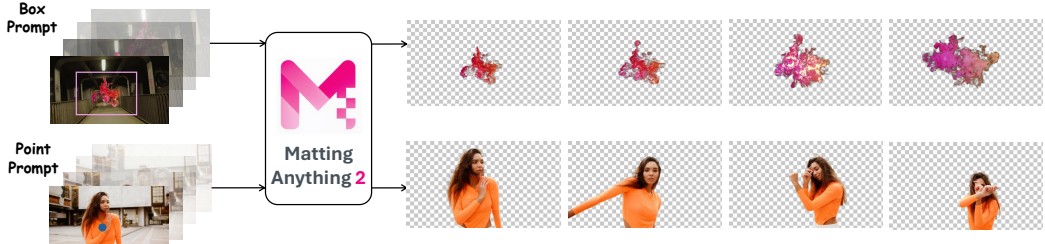

Figure 1: Matting Anything 2 is a video model that can be directly driven by user prompts and is able to process diverse objects in addition to human portraits.

Table 1: Statistics of popular video matting benchmarks. Count is calculated based on the number of distinguished foregrounds. Average Duration refers to the average frame count of the clips. Domain refers to the categories of objects.

| Test Set | Number | Average Duration | Domain |
|---|---|---|---|
| VideoMatte240K Lin et al. (2021b) | 25 | 100 | human |
| VideoMatting108 Zhang et al. (2021b) | 28 | 845 | human, cloth, smoke |
| YoutubeMatte Yang et al. (2025) | 32 | 100 | human |
| Natural Video Matting | 50 | 164 | animals, bubble, cloud, fire, water, frost, plant... |

To address these challenges, we introduce Matting Anything 2 (MAM2), a video matting model that can handle diverse objects. We developed MAM2 by building upon the foundation of SAM2 Ravi et al. (2024), thereby inheriting its excellent interactive capabilities. This allows MAM2 to accept various forms of interaction, including points, boxes, and masks. Furthermore, we follow the paradigm of recent video matting models, which requires user interaction only on the first frame, minimizing the user's interaction cost as much as possible, as shown in Fig. 1.

To endow MAM2 with strong generalization capabilities, we propose the Promptable Dual-mode Decoder (PDD). In contrast to the original mask decoder of SAM2, PDD can simultaneously predict an object's segmentation mask and its corresponding trimap, which serves as strong guidance for the final alpha matte prediction. We adopted this technical approach, motivated by the demonstrated dominance of trimap-based methods Dosovitskiy et al. (2020); Hu et al. (2024) in the field of natural image matting. By strengthening the guidance from the mask for the trimap prediction, PDD is able to generate high-quality trimaps for common objects.

However, we found that simply using PDD to predict per-frame trimaps for transparent objects results in unstable quality. This issue is primarily caused by the decoding mechanism of SAM2. For frames without a user prompt (i.e., all frames after the first), the decoding process relies mainly on embedding the mask memory from the previous frame into the image features. Yet, for transparent objects, decoding a trimap based on mask memory is particularly challenging. This is because large transparent areas require the prediction of more *unknown* regions in the trimap rather than *foreground* regions. This, in turn, increases the discrepancy between the trimap and the mask, which means that the disparity between the ideal features required for their respective decoding also grows. To resolve this, we propose the Memory-Separable Siamese (MSS) mechanism. MSS employs a recurrent approach to bypass the interference that mask memory causes during trimap decoding: after the segmentation mask is predicted by PDD, this mask is used as a prompt to drive the PDD a second time to generate the trimap. Crucially, this second decoding pass utilizes pre-saved image features that have not undergone the mask memory embedding. Experiments show that MSS significantly improves the stability of trimap predictions for transparent objects. Furthermore, since the two passes share parameters, this siamese architecture adds no additional parameters.

To validate the generalization performance of our method on natural scenes, we introduce a new, advanced benchmark: the Natural Object Video Matting (NOVM) dataset. In contrast to existing video matting test sets, NOVM exhibits significantly greater domain diversity, encompassing categories beyond portraits such as plants, fire, water, and more, as shown in Table 1.

We conducted extensive experiments on both existing human portrait benchmarks and our newly proposed NOVM dataset. Compared to the state-of-the-art model, MAM2 reduces the MAD from 39.44 to 14.72 on NOVM (Natural Object Video Matting) and from 2.05 to 1.16 on Youtube-Matte Yang et al. (2025) (human video matting). Therefore, MAM2 is not a model specifically optimized for transparent objects; it also outperforms existing methods in human matting. Furthermore, MAM2 is also a powerful image matting method. We also evaluate its performance on image matting tasks, where it achieves competitive results. We believe that Matting Anything 2 holds immense value for practical applications.

## 2 RELATED WORKS

### 2.1 VIDEO OBJECT SEGMENTATION

Video Object Segmentation (VOS) tasks are primarily divided into several categories: unsupervised VOS, semi-supervised VOS, referring VOS Liang et al. (2025); Cuttano et al. (2025); Li et al. (2023b) and interactive VOS. Unsupervised VOS Lee et al. (2023); Li et al. (2024c); Cho et al. (2024); Zhuge et al. (2024) does not require any user-provided annotations for guidance, allowing the model to perform segmentation automatically. However, this type of methods suffers from the inability to specify a target object and often exhibits lower accuracy and consistency.

Consequently, plenty of work has been dedicated to semi-supervised VOS methods Caelles et al. (2017). These methods require the user to provide an object mask for the first frame, and the model then segments the object in all subsequent frames based on this initial mask. Numerous classic architectures have been proposed to address this task. These include mask propagation-based methods Oh et al. (2018); Garg & Goel (2021), which use the mask from the previous frame as guidance for the current one to achieve coherent segmentation throughout the sequence, and memory-based methods Oh et al. (2019); Cheng & Schwing (2022); Zhou et al. (2024), which rely on feature matching between the current and historical frames to ensure temporal consistency.

Recently, the introduction of SAM2 has drawn significant attention to interactive VOS methods. This category of approaches aims to achieve video segmentation through more user-friendly interactions, such as clicks. Moreover, users can refine the segmentation results based on the model's predictions. In particular, SAM2's compatibility with multiple prompt types and its strong generalization capabilities have inspired a considerable amount of subsequent work Ding et al. (2024); Cuttano et al. (2025); Yang et al. (2024).

### 2.2 VIDEO MATTING

Video Matting methods can similarly be categorized based on the type of user guidance they require. Automatic video matting models Ke et al. (2022); Lin et al. (2022), can predict precise alpha mattes without any user input, but they lack the ability to specify a target object. In contrast, approaches that follow the VOS paradigm Yang et al. (2025); Huynh et al. (2024) require a user-provided mask in the first frame to select the target for matting. However, both of these method categories are typically limited to processing human or animal portraits.

Alternatively, other methods utilize a trimap instead of a mask of the first frame for guidance Seong et al. (2022); Huang & Lee (2023). Guided by the stronger prior information provided by a trimap, these methods not only achieve high matting accuracy but have also shown the potential to handle objects beyond portraits. However, it is evident that the user interaction cost of providing a trimap is significantly higher than that of a mask.

Additionally, there is a special type of background-based video matting methods Lin et al. (2021a); Sengupta et al. (2020); Xu et al. (2021). These methods require the user to provide an image of clean background without the foreground object, as auxiliary information to achieve precise matting. However, in practical applications, a clean background is often unavailable. Furthermore, these methods impose strict requirements on background consistency, making them unable to adapt to temporal changes. Therefore, compared to the aforementioned methods, background-based matting is less practical to deploy in real-world scenarios.

# 3 METHODOLOGY

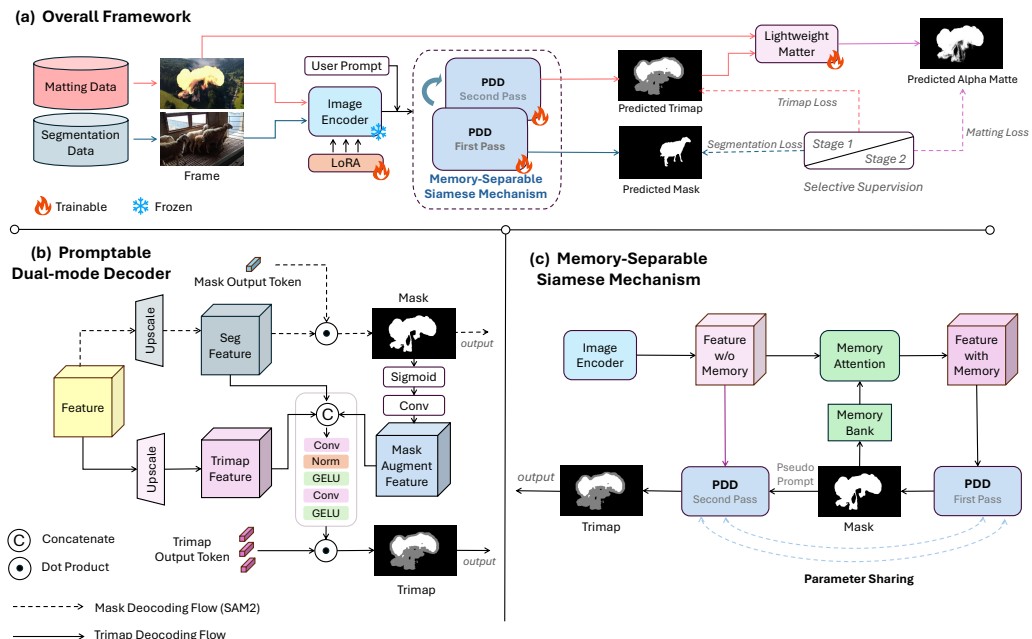

Figure 2: Architecture of Matting Anything 2. MAM2 first predicts the object's trimap based on the user's prompt, and then uses the trimap to predict the final alpha matte.

## 3.1 OVERALL FRAMEWORK

As shown in Fig. 1, MAM2 is capable of predicting alpha mattes for an entire video, requiring user interaction on only the first frame. Users can provide prompts directly to MAM2, eliminating the need for an auxiliary interactive segmentation model to generate an mask for the first frame. Furthermore, MAM2 inherits the strong interactivity of SAM2, allowing it to support various prompt types such as boxes, points, and masks, which ensures high user-friendliness.

We implement this inference process using a progressive decoding pipeline. As depicted in Fig. 2, the pipeline begins with an image encoder of SAM2 finetuned by LoRA that extracts the image feature of a video frame. Subsequently, the Memory-Separable Siamese (MSS) mechanism utilizes two sequential Promptable Dual-mode Decoder (PDD) passes to predict the target's mask and trimap based on the user prompt. Finally, a lightweight trimap-based matting module predicts the final alpha matte. MAM2 is compatible with trimap-based matte model of various architectures. We choose MEMatte Lin et al. (2025) as our lightweight matter here due to its efficiency, with details provided in A.1

## 3.2 SELECTIVE SUPERVISION SCHEME

The task of Video Matting (VM) suffers from significant data scarcity, compelling most approaches Yang et al. (2025); Huang & Lee (2023) to supplement their training data with Image Matting (IM) or Video Object Segmentation (VOS) data. Thus, we propose a selective supervision scheme to better utilize the knowledge contained within these heterogeneous data sources. The implementation of this strategy is enabled by MAM2's capability to concurrently generate multiple outputs, including the mask, trimap, and alpha matte.

To facilitate this specialized learning, we partition the training procedure into two stages. Let the model parameters $\Theta$ be partitioned into $\theta_{\text{main}}$ for the main components (encompassing all parameters excluding the lightweight matter module) and $\theta_{\text{matter}}$ for the lightweight matter module.

The first stage is designed to optimize $\theta_{\text{main}}$ using data from IM, VM, and VOS. Accordingly, the loss for this main stage, $L_{\text{main}}$, is formulated as follows:

$$L_{\text{main}} = \mathbb{I}_{\text{VOS}} \cdot \mathcal{L}_{\text{mask}}(M, y_M) + (\mathbb{I}_{\text{VM}} + \mathbb{I}_{\text{IM}}) \cdot \mathcal{L}_{\text{trimap}}(T, y_T) \tag{1}$$

where $M$ and $T$ are the mask and trimap outputs, with corresponding ground truths $y_M$ and $y_T$. The indicator functions $\mathbb{I}_{\text{VOS}}$, $\mathbb{I}_{\text{VM}}$, and $\mathbb{I}_{\text{IM}}$ selectively activate the appropriate loss term based on the type of data randomly loaded by the dataloader from the mixed training set. This stage trains MAM2 to robustly produce a coarse mask or a trimap.

In the second stage, only the lightweight matter parameters $\theta_{\text{matter}}$ are optimized. For this stage, we define a separate loss, $L_{\text{matter}}$:

$$L_{\text{matter}} = \mathbb{I}_{\text{IM}} \cdot \mathcal{L}_{alpha}(\alpha, y_\alpha) \tag{2}$$

where $\alpha$ is the final predicted alpha matte and $y_\alpha$ is its ground truth. This stage exclusively uses image matting data for supervision. This is because this stage focuses on learning fine-grained detail perception, which demands high-fidelity annotations. Notably, the annotation quality of image matting data is significantly superior to that of video matting data, an observation also noted in MatAnyone Yang et al. (2025).

Further details concerning the assignment of specific data sources to the optimization of different model parameters can be found in Appendix A.2. The composition of each category of datasets can be found in Appendix A.3. The detailed formulation of each loss function and the sampling strategy for the training data are provided in Appendix A.5.

## 3.3 PRELIMINARY

To facilitate a better understanding of our method, we begin by providing a brief preliminary on the decoding mechanism of SAM1&2 Kirillov et al. (2023); Ravi et al. (2024) before detailing the model architecture of MAM2. To ensure consistency with the task setting of this paper, all subsequent descriptions are based on the standard semi-supervised VOS setting, where the user provides a prompt only for the first frame of a video sequence.

As SAM1 is an image segmentation model that requires user-provided prompts, it utilizes a Promptable Mask Decoder. This decoder takes the features extracted by image encoder and the user's prompt as input to predict the segmentation mask. This process can be formally formulated as:

$$M = f_{\text{Decoder}}(\mathbf{F}, \mathcal{P}_{\text{user}}) \tag{3}$$

where $M$ is the predicted segmentation mask, $\mathbf{F}$ represents the image feature, and $\mathcal{P}_{\text{user}}$ is the user-provided prompt.

However, since SAM2 is required to predict masks for every frame based on sparse prompts, a memory mechanism is introduced to address this challenge. For subsequent frames, which lacks a user prompt, the memory mechanism embeds the mask prediction from the previous frame into the current image features via memory attention. This embedded memory functions as an implicit prompt, replacing the absent user prompt and driving the Promptable Mask Decoder's operation. The per-frame decoding process of SAM2 can be formally described as:

$$M^t = \begin{cases} f_{\text{Decoder}}(\mathbf{F}^t_{\text{non-mem}}, \mathcal{P}_{\text{user}}), & \text{if } t = 0 \\ f_{\text{Decoder}}(\mathbf{F}^t_{\text{mem}}, \emptyset), & \text{if } t > 0 \end{cases} \tag{4}$$

where $t$ is the frame index, $M^t$ is the mask prediction at frame $t$, $\mathbf{F}^t_{\text{non-mem}}$ represents the image feature of frame $t$ without memory embedding, $\mathbf{F}^t_{\text{mem}}$ denotes the image feature of frame $t$ embedded with memory of mask predictions of previous frames, and $\emptyset$ represents the absent user prompt.

## 3.4 PROMPTABLE DUAL-MODE DECODER

MAM2 operates as a progressive decoding pipeline; consequently, it necessitates the preliminary prediction of the target's trimap to facilitate the subsequent prediction of the final alpha matte. To address this requirement, we propose the Promptable Dual-mode Decoder (PDD). Distinct from the original SAM2 decoder, the PDD is capable of simultaneously predicting both a segmentation mask

and a trimap for every frame. We denote the PDD as the function $f_{\text{PDD}}$. Similar to Equation 4, $f_{\text{PDD}}$ can be described as:

$$(M^t, T^t) = \begin{cases} f_{\text{PDD}}(\mathbf{F}^t_{\text{non-mem}}, \mathcal{P}_{\text{user}}), & \text{if } t = 0 \\ f_{\text{PDD}}(\mathbf{F}^t_{\text{mem}}, \emptyset), & \text{if } t > 0 \end{cases} \tag{5}$$

where $M^t$ and $T^t$ represent the mask and trimap prediction at frame $t$, respectively.

However, predicting an additional trimap based on the original SAM2 decoder is non-trivial. While a line of work has focused on refining the coarse masks predicted by SAM1 into finer and more detailed ones Ke et al. (2023); Liu et al. (2024), these masks, whether coarse or fine, remain semantically consistent. The trimap we aim to predict, however, is fundamentally a different form of semantic representation. To address this distinction, SEMatte Xia et al. (2024), a SAM-based image matting method, proposes employing a fully independent parallel branch prior to the final decoding to predict this distinct representation. However, our findings indicate that while a simple parallel structure facilitates the simultaneous prediction of masks and trimaps, the resulting trimaps are often noisy, frequently exhibiting jagged artifacts along the boundaries, as illustrated in the second row of Fig. 3.

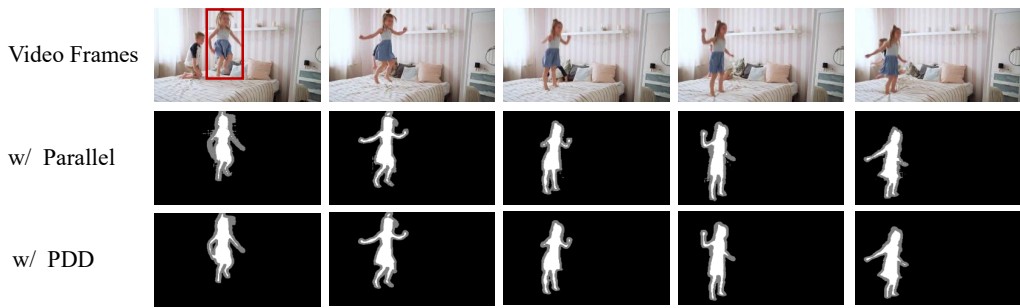

Figure 3: PDD improves the quality of the trimap prediction. Zoom in to observe details.

We attribute this instability to the fact that a simple parallel branch fails to fully leverage SAM's robust semantic understanding and the exceptional stability of its mask predictions. Therefore, we consider it essential to incorporate strong guidance from SAM2's mask predictions into the parallel branch. Motivated by this, we propose the trimap decoding flow in PDD, as illustrated in part (b) of Fig. 2.

First, the predicted mask is normalized with a sigmoid function and then processed by a convolutional layer to generate a *mask augment feature*. Then, the *mask augment feature* is concatenated with the original *segmentation feature* and *trimap feature*. Subsequently, a lightweight fusion module performs a mask-guided enhancement on these concatenated features. Finally, the final trimap is obtained by computing the dot product between the fused features and the *trimap output token*. As illustrated in Fig. 3, this simple design leads to a significant improvement in the stability and quality of the predicted trimap. This, in turn, enhances the quality of the final predicted alpha matte. This simple design yields improvements of 24% and 29% on the natural object and human portrait benchmarks, respectively, as shown in the top two rows of Table 5.

In addition, PDD inherits sam2's excellent compatibility with multiple prompt types, which makes the implementation of Memory-Separable Siamese Mechanism possible, as will be discussed in detail in the next section.

### 3.5 MEMORY-SEPARABLE SIAMESE MECHANISM

A "matting anything" method must do more than just competently handle portraits or objects where fine details are concentrated at the boundary. A true test of its capability lies in processing challenging transparent objects like fire and bubbles, which are characterized by extensive transparency and complex details.

When we applied our PDD-equipped MAM2 to these objects, we discovered a strange temporal collapse. While the trimap for the first frame is predicted accurately, from the second frame onward, we found that *unknown* regions in the trimap were prone to being misclassified as *foreground*, as

shown in Fig. 4. For our lightweight matter, which relies on the trimap predicted by PDD to predict the final alpha matte, such false positive errors of *foreground* are known to be catastrophic, particularly for objects with large-scale transparency.

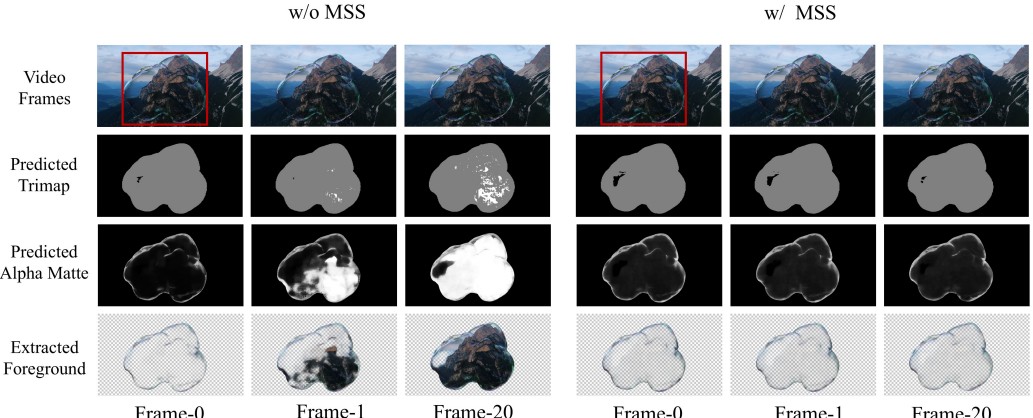

Figure 4: The entire region of the bubble should be predicted as *unknown* in the trimap. MSS effectively mitigates the collapse of trimap prediction for subsequent frames when handling transparent objects.

As discussed in Section 3.3 and Equation 5, the decoding process for the first frame is driven by the user prompt, whereas for subsequent frames, it relies on the memory of the previous mask prediction embedded within the image features. Consequently, this anomalous temporal collapse can be attributed to a specific disparity: while PDD performs accurate trimap decoding when guided by user prompts, it fails to do so when guided by the memory embedded in the image features. Therefore, we identify the root cause as the process of embedding the memory of previous mask predictions into the image features. We posit that this embedding operation induces a significant shift in the image feature space, which severely interferes with the trimap decoding process. Corroborating evidence for this is observed in the erroneous trimap predictions of subsequent frames, which exhibit a tendency to resemble binary segmentation masks rather than trimaps. Notably, these predictions display a strong bias towards classifying object regions—regardless of their actual transparency—as definite *foreground* rather than the *unknown* category required for trimaps.

We consider this phenomenon to be logical. Given that masks and trimaps represent distinct semantic concepts, an inconsistency between the ideal feature spaces required for their respective decoding is to be expected. In essence, while subsequent frames rely on mask memory for decoding, the embedding of this memory simultaneously interferes with trimap prediction.

To overcome this dilemma, we introduce the Memory-Separable Siamese Mechanism (MSS), a recurrent approach to trimap decoding. For subsequent frames, once PDD computes a segmentation mask, we leverage this mask as a pseudo-prompt to drive a second PDD pass. Crucially, this second pass is performed on a preserved, memory-free version of the image feature that was saved prior to the memory attention.

Consequently, the decoding of subsequent frames is effectively realigned to rely on memory-free features and a "user prompt", thereby avoiding the interference caused by the shifted image feature on trimap decoding, as indicated by the purple connecting line in part (c) in Fig. 2. The trimap generated from this second pass is then selected as the final output. This process can be formally formulated as:

$$M^t = \pi_1\left(f_{\text{PDD}}(\mathbf{F}_{\text{mem}}^t, \emptyset)\right) \tag{6}$$

$$T^t = \pi_2\left(f_{\text{PDD}}(\mathbf{F}_{\text{non-mem}}^t, M^t)\right) \tag{7}$$

where $\pi_x$ denotes the projection function that extracts the $x$-th element from the output tuple (e.g., $\pi_1$ extracts the mask and $\pi_2$ extracts the trimap).

As illustrated in Fig. 4, MSS substantially enhances the robustness of MAM2 on challenging objects characterized by large-scale transparency. Furthermore, since the mask used to drive the second pass

is decoded from features with memory, temporal consistency of the trimap can be transmitted and maintained by this mask. In addition, since the PDD weights are shared between two passes, MSS is a siamese architecture, adding no additional parameters. Meanwhile, because PDD is built upon SAM's lightweight mask decoder, the computational overhead of this second pass is negligible.

## 4 EXPERIMENTS

### 4.1 IMPLEMENTATION DETAILS

Consistent with recent video matting methods Yang et al. (2025), MAM2 is trained on multiple types of data: video object segmentation Ding et al. (2023), video matting Zhang et al. (2021b), image segmentation Qin et al. (2022), and image matting Xu et al. (2017); Qiao et al. (2020); Li et al. (2022); Ma et al. (2023); Cai et al. (2022). No private or proprietary data was used to train MAM2. The entire training set is composed of publicly accessible datasets. The specific datasets used for training are detailed in Appendix A.3.

$\theta_{\mathrm{main}}$ are finetuned from pretrained SAM2 weights for 100 epochs using the AdamW optimizer with an initial learning rate of $4 \times 10^{-4}$. The LoRA rank for the encoder is set to 16. $\theta_{\mathrm{matter}}$, are trained from a ViT-Small model initialized with DINO pretrained weights. This component is trained for approximately 3,500 iterations using AdamW with an initial learning rate of $5 \times 10^{-4}$. The total number of trainable parameters is 44.7M. Further training details are available in the Appendix A.

### 4.2 NATURAL OBJECT VIDEO MATTING DATASET

To evaluate the performance of methods on diverse objects, we introduce Natural Object Video Matting (NOVM), the first video matting benchmark composed of a rich variety of natural objects. The construction of NOVM began with the collection of After Effects assets that included preexisting alpha mattes, allowing them to be used directly as assets in content creation. Subsequently, this collection was filtered to discard assets with insufficient detail, as well as all cartoon-styled clips, which we noted constituted a significant portion of the initial set. Finally, the retained assets are composited onto high-resolution and dynamic backgrounds to produce the final video clips and corresponding alpha matte clips.

The final NOVM dataset contains 50 clips, each featuring a distinct foreground object or action and a dynamic background. Most importantly, NOVM covers a highly diverse range of object domains, presenting a formidable challenge to existing video matting models. We provide several examples of NOVM in Fig.7 and the breakdown of NOVM in Table 9.

### 4.3 VIDEO MATTING

Table 2: Quantitative comparison with other video matting methods in interactive mode on NOVM and YoutubeMatte Datasets.

| Method | Prompt | NOVM (natural objects) | | | | YoutubeMatte (human) | | | |
|---|---|---|---|---|---|---|---|---|---|
| | | MAD ↓ | MSE ↓ | GRAD ↓ | dtSSD ↓ | MAD ↓ | MSE ↓ | GRAD ↓ | dtSSD ↓ |
| TCVOM Zhang et al. (2021a) | Trimap | 56.18 | 38.90 | 153.95 | 3.84 | 1.57 | 0.40 | 6.74 | 1.52 |
| FTP-VM Huang & Lee (2023) | Trimap | 37.98 | 19.90 | 78.06 | 4.24 | 2.26 | 1.10 | 5.63 | 1.70 |
| MaGGIe Huynh et al. (2024) | Mask | 50.04 | 35.23 | 108.01 | 4.90 | 2.37 | 0.98 | 7.69 | 1.77 |
| MatAnyone Yang et al. (2025) | Mask | 39.44 | 25.63 | 89.60 | 4.10 | 2.05 | 0.76 | 9.67 | 1.75 |
| **Matting Anything 2 (Ours)** | Mask | 15.19 | 4.27 | 26.45 | 2.80 | 1.16 | 0.24 | 3.12 | 1.21 |
| **Matting Anything 2 (Ours)** | Box & Point | **14.72** | **3.70** | **23.54** | **2.65** | **1.16** | **0.24** | **3.07** | **1.20** |

We first compare the performance of the methods when prompted by a mask of the first frame, following the setting of semi-supervised VOS task, as presented in Table 2. We select NOVM and YoutubeMatte Yang et al. (2025) as our test datasets. The former is used to evaluate the model's performance on diverse objects, while the latter assesses its performance on typical human portraits. MAM2 demonstrates a significant advantage across all metrics, including MAD and MSE for overall prediction, Grad for detail fidelity, and dtSSD for temporal consistency. More importantly, MAM2 can even perform matting without the need for a user-provided mask; it can be driven directly by

user-provided points or boxes. Even in this mode, MAM2 continues to exhibit exceptionally strong performance.

Following the MatAnyone Yang et al. (2025), we also evaluated the performance of MAM2 in an automatic human matting mode. This mode operates by using an automatic human matting model to obtain the matte for the first frame. For a fair comparison, both MAM2 and MatAnyone utilize RVM to generate this first-frame matte. As shown in Table 3, MAM2 also demonstrates the strongest overall performance in automatic matting.

Table 3: Quantitative comparison with other video matting methods in automatic mode on Youtube-Matte and VM240K Datasets.

| Method | YoutubeMatte (human) | | | | VM240K (human) | | | |
|---|---|---|---|---|---|---|---|---|
| | MAD↓ | MSE↓ | GRAD↓ | dtSSD↓ | MAD↓ | MSE↓ | GRAD↓ | dtSSD↓ |
| MODNet Ke et al. (2022) | 15.29 | 12.68 | 8.42 | 2.74 | 11.13 | 5.54 | 15.30 | 3.08 |
| RVM Lin et al. (2022) | 4.37 | 2.25 | 15.1 | 2.28 | 6.57 | 1.93 | 10.55 | 1.90 |
| RVM-Large Lin et al. (2022) | 3.50 | 1.19 | 12.64 | 2.08 | 5.81 | **0.97** | 9.65 | 1.78 |
| MatAnyone Yang et al. (2025) | 3.70 | 2.35 | 11.45 | 1.81 | 5.66 | 1.68 | 5.75 | 1.27 |
| **Matting Anything 2 (Ours)** | **1.19** | **0.27** | **3.17** | **1.23** | **5.10** | 1.10 | **4.15** | **1.26** |

## 4.4 IMAGE MATTING

Recently, a wave of image matting methods built upon the foundation of SAM 1/2 has emerged, including ZIM Kim et al. (2025), Matting Anything Li et al. (2023a), and SEMatte Xia et al. (2024). MAM2 not only shows strong performance in video matting but is also a powerful image matting model. For image matting, MAM2 can also be driven by efficient user prompts such as points and boxes. We provide a comparison with other image matting methods in Table 4. Specifically, in addition to a visual prompt, SDMatte requires a flag to specify whether the matting target is a transparent object. Therefore, to ensure a fair comparison with other methods, we report the results for SDMatte without this text prompt. We provide visualization results of image matting in Appendix C.2. Notably, the model parameters of MAM2 for the video matting and image matting tasks are identical.

Table 4: Quantitative comparison of image matting methods with various prompt types on AIM-500. N/A indicates that the model does not support Click mode. The * denotes a different version of SDMatte.

| Method | Prompt | MSE | SAD | Grad | Conn | Prompt | MSE | SAD | Grad | Conn |
|---|---|---|---|---|---|---|---|---|---|---|
| Matting Anything Li et al. (2024b) | Box | 11.60 | 36.66 | 21.04 | 18.99 | Point | 7.52 | 186.50 | 37.48 | 40.38 |
| SmartMatting Ye et al. (2024) | Box | 7.65 | 25.33 | 27.16 | 13.54 | Point | 30.20 | 66.27 | 46.63 | 18.77 |
| SEMatte Xia et al. (2024) | Box | 7.65 | 24.30 | 16.06 | 13.64 | Point | N/A | N/A | N/A | N/A |
| SDMatte Huang et al. (2025) | Box | 10.04 | 29.35 | 24.06 | 15.62 | Point | 11.93 | 33.57 | 29.15 | 18.15 |
| SDMatte* Huang et al. (2025) | Box | 4.91 | 19.81 | 15.84 | 11.97 | Point | N/A | N/A | N/A | N/A |
| **Matting Anything 2 (Ours)** | Box | **4.24** | **18.07** | **13.88** | **11.01** | Point | **5.68** | **20.78** | **14.63** | **10.90** |

## 4.5 ABLATION STUDY

We present the ablation studies for PDD and MSS here. The first two rows of Table 5 represent the ablation study for PDD, where *Parallel* refers to the simple parallel structure for trimap prediction, as described in Sec 3.4. In conjunction with Fig. 3, it is evident that PDD substantially improves the stability of the predicted trimap, thereby enhancing the final matting quality.

Rows 2, 3, and 4 detail the ablation study for MSS. In this context, MCS (Memory-Consistent Siamese) represents the strategy that simply performs a second pass of PDD based on memory-embedded image features. The results indicate that the critical factor for MSS's significant improvement is not an extra decoding pass (as evidenced by the lack of improvement from MCS). Instead, the key is the utilization of memory-free image features during the second PDD pass, which prevents interference from mask memory in the trimap decoding process.

Table 5: Ablation study of different components of Matting Anything 2.

| Parallel | PDD | MCS | MSS | NOVM | | | | Youtube | | | |
|---|---|---|---|---|---|---|---|---|---|---|---|
| | | | | MAD ↓ | MSE ↓ | GRAD ↓ | dtSSD ↓ | MAD ↓ | MSE ↓ | GRAD ↓ | dtSSD ↓ |
| ✓ | | | | 26.19 | 13.21 | 43.29 | 3.14 | 1.54 | 0.52 | 3.49 | 1.30 |
| | ✓ | | | 18.55 | 6.77 | 29.77 | 2.91 | 1.16 | 0.24 | 3.08 | 1.19 |
| | ✓ | ✓ | | 20.23 | 8.59 | 39.26 | 2.85 | 1.18 | 0.26 | 3.14 | 1.19 |
| | ✓ | | ✓ | 14.72 | 3.70 | 23.54 | 2.65 | 1.16 | 0.24 | 3.07 | 1.20 |

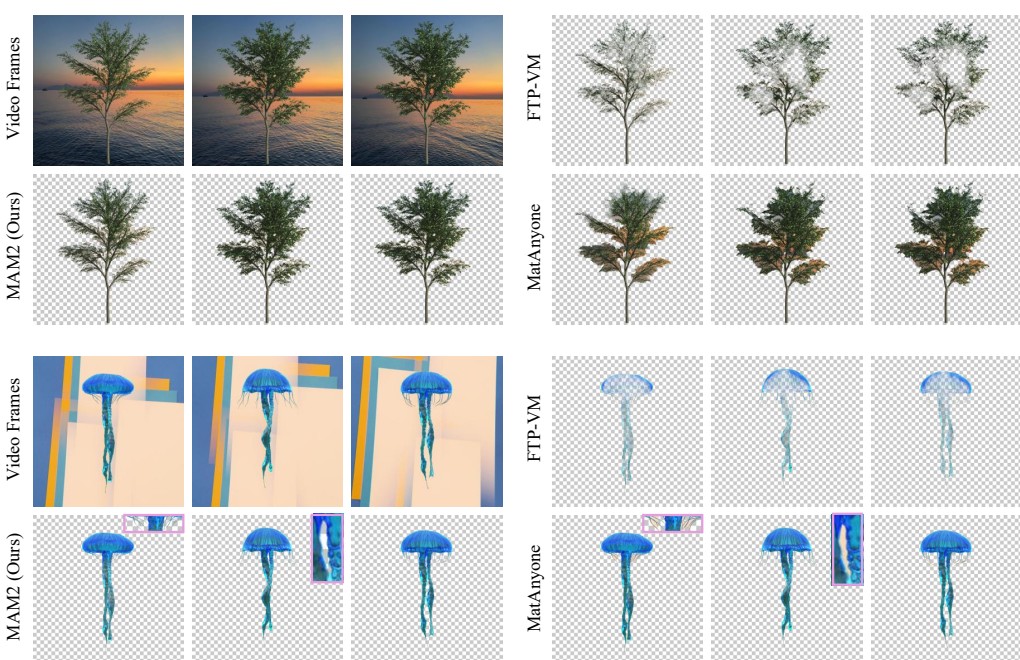

Figure 5: Visual comparison with other video matting methods. Zoom in to observe details.

## 4.6 VISUALIZATION

In Fig. 5, we provide a visual comparison of MAM2 with MatAnyone Yang et al. (2025) and FTP-VM Huang & Lee (2023), which represent the state-of-the-art in mask-guided and trimap-guided video matting, respectively. MAM2 provides higher-quality matting results even when driven directly by user-provided points and boxes.

We provide extensive visualization results in Appendix C.1. Specifically, Figs. 8 and 9 demonstrate performance on the NOVM dataset, Fig. 10 on the YouTubeMatte dataset, and Figs. 11 and 12 on real-world videos. We also provide several results in MP4 format in the supplementary material.

## 5 CONCLUSION

In this paper, we introduce Matting Anything 2, a powerful model designed for matting any object in videos. We propose a Promptable Dual-mode Decoder to enable a seamless, interactive workflow and a Memory-Separable Siamese mechanism to enhance generalization for complex objects by resolving memory conflicts without adding parameters. To facilitate robust evaluation, we also present the Natural Object Video Matting dataset, a new benchmark with significant domain diversity. Experimental results demonstrate that MAM2 significantly outperforms existing methods, establishing a new state-of-the-art for both natural and portrait scenes. We believe this work holds immense potential for significant practical applications.

ACKNOWLEDGMENTS

This work was supported by the National Natural Science Foundation of China (No. 62476124, 62172090), Fundamental and Interdisciplinary Disciplines Breakthrough Plan of the Ministry of Education of China (JYB2025XDXM902), Natural Science Foundation of Jiangsu Province (No. BK20242015, BK20230833), and the Gusu Innovation and Entrepreneur Leading Talents (No. ZXL2025322).

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

## A IMPLEMENTATION DETAILS

### A.1 LIGHTWEIGHT MATTER

We employ a lightweight trimap-based matter, MEMatte Lin et al. (2025), to predict the alpha matte. This matter integrates the trimap predicted by PDD with the original RGB image to produce the final alpha matte.

MEMatte is a standard trimap-based matting method bsaed on a ViT Dosovitskiy et al. (2020) backbone. Its core advantage lies in an adaptive token routing mechanism that drastically reduces the number of tokens participating in the computation within the global attention blocks of its ViT backbone. Because the computational complexity of the attention mechanism scales quadratically with the number of tokens, MEMatte can significantly lower the memory usage and latency during inference.

The target compression degree is set to 0.25, with a maximum token number of 12,000. For the router and lightweight token refinement module, the linear layers are initialized with a truncated normal distribution (std=0.02), while the LayerNorm layers have biases set to zero and weights set to one. The remaining modules are initialized with the teacher model.

During training, data augmentations include random affine transformations, random cropping, random jitter, random horizontal flipping, and composition, among others. The inputs are randomly cropped into $1024 \times 1024$ patches. Essentially, MAM2 retains the same model architecture as MEMatte, with the only modification being to the training data.

### A.2 SELECTIVE SUPERVISION SCHEME

As illustrated in Fig. 6, we employ a two-stage training pipeline utilizing a Selective Supervision Scheme.

During *Stage 1*, we deploy a mixed dataloader that randomly samples training batches from heterogeneous sources—specifically, Segmentation Data and Matting Data. This stage focuses on optimizing the main model parameters, $\Theta_{\text{main}}$, with the supervision signal adapting dynamically to the sampled data type. For Segmentation Data, the model is supervised via the Segmentation Loss on the predicted mask. Conversely, when Matting Data is sampled, the model minimizes the Trimap Loss. The ground truth trimap required for this supervision is generated by degrading the ground truth alpha matte through standard dilation, erosion, and quantization operations. This approach is a well-established technique in image matting for synthetically generating trimaps.

Subsequently, *Stage 2* exclusively optimizes the lightweight matter parameters, $\Theta_{\text{matter}}$, loading only Matting Data to refine the final alpha matte under the guidance of the Matting Loss.

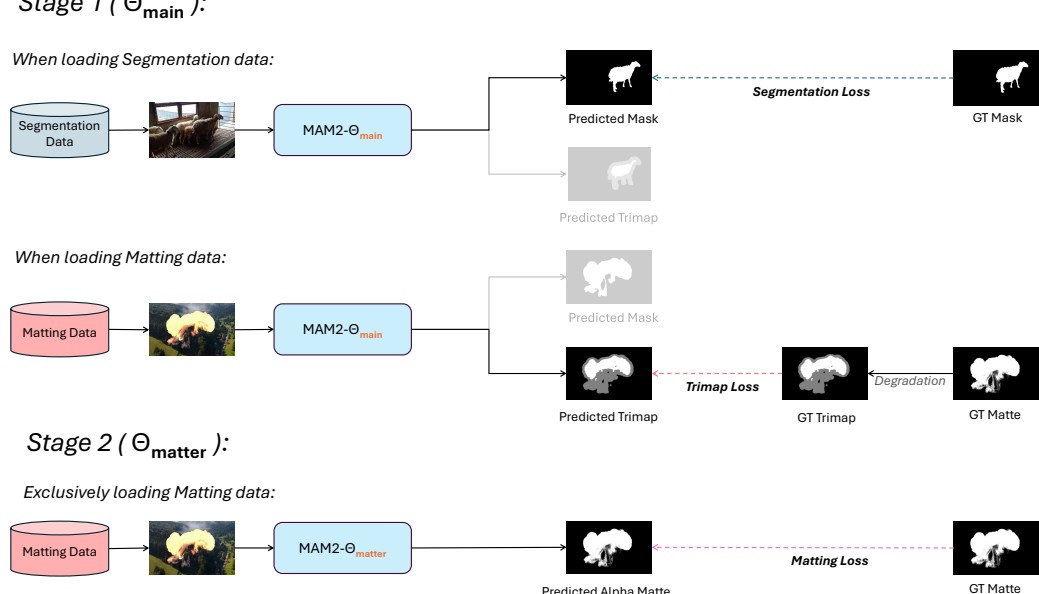

Figure 6: Pipeline of Selective Supervision Scheme.

## A.3  TRAINING DATA

The field of video matting has long faced the issue of data scarcity. Consequently, individual methods employ their own strategies to augment training data by sourcing it from other fields, such as video segmentation, image segmentation, image matting. As the field has evolved, the datasets used by different methods have diverged, making it exceedingly difficult to ensure identical training data settings for a fair comparison. We can ensure that MAM2 is trained exclusively on available public datasets. We list the composition of the training data for recent video matting methods in Table 6. It should be noted that the relatively small image counts for P10K, AM-2K, DIS5K, and DUTS are due to the fact that MAM2 utilizes only a subset of these datasets for training, rather than the full sets. For instance, while the P10K training set contains 9,421 images, MAM2 employs only 1,000 of them.

| Methods | Total Count | | Video Segmentation | | Video Matting | | | Image Segmentation | | | | Image Matting | | | | | |
|---|---|---|---|---|---|---|---|---|---|---|---|---|---|---|---|---|---|
| | Images | Videos | YoutubeVIS | MOSE | VM800 | VM108 | VM240K | COCO | SPD | DUTS | DIS5K | DIM | D-646 | AM-2K | T-460 | I-HIM50K | P10K |
| FTP-VM | 596 | 3551 | ✓ | | | ✓ | | | | | | | ✓ | | | | |
| MaGGIe | 49373 | 555 | | | | ✓ | ✓ | | | | | | | | | ✓ | |
| MatAnyone | 89251 | 4297 | ✓ | | ✓ | | | ✓ | ✓ | | | ✓ | ✓ | ✓ | ✓ | | |
| MAM2 | 12239 | 1326 | | ✓ | | ✓ | | | | ✓ | ✓ | ✓ | ✓ | ✓ | ✓ | | ✓ |

Table 6: Composition of training datasets for different methods.

We have summarized the sampling probabilities for each dataset type, along with other fundamental configurations, in Table 7. Num Frames denotes the number of frames loaded at once; this setting is only applicable to video data.

|  | Sample Probability | Batch Size | Num Frames |
|---|---|---|---|
| Image Segmentation | 0.25 | 4 | - |
| Image Matting | 0.25 | 4 | - |
| Video Segmentation | 0.25 | 1 | 4 |
| Video Matting | 0.25 | 1 | 4 |

Table 7: Settings for training with multiple datasets.

## A.4 ABLATION OF TRAINING SETS

As shown in Table 6, recent matting methods utilize diverse training configurations. Due to the non-public nature of certain datasets and training codes, it is infeasible to align the training settings across all methods. However, to demonstrate that the performance improvements of our method are not derived from our specific training set, we aligned our training data with that of FTP-VM and retrained MAM2. As evidenced in Table 8, MAM2 continues to exhibit significantly superior performance. We selected FTP-VM for this validation because it employs the most straightforward training data composition, consisting only of VM108, YouTubeVIS, and D-646.

Table 8: Ablation study of different training sets.

| Method | Training Data | NOVM (natural objects) | | | | Youtube (human) | | | |
|---|---|---|---|---|---|---|---|---|---|
|  |  | MAD ↓ | MSE ↓ | GRAD ↓ | dtSSD ↓ | MAD ↓ | MSE ↓ | GRAD ↓ | dtSSD ↓ |
| TCVOM Zhang et al. (2021a) | Respective Sets | 56.18 | 38.90 | 153.95 | 3.84 | 1.57 | 0.40 | 6.74 | 1.52 |
| FTP-VM Huang & Lee (2023) | Respective Sets | 37.98 | 19.90 | 78.06 | 4.24 | 2.26 | 1.10 | 5.63 | 1.70 |
| MaGGIe Huynh et al. (2024) | Respective Sets | 50.04 | 35.23 | 108.01 | 4.90 | 2.37 | 0.98 | 7.69 | 1.77 |
| MatAnyone Yang et al. (2025) | Respective Sets | 39.44 | 25.63 | 89.60 | 4.10 | 2.05 | 0.76 | 9.67 | 1.75 |
| **Matting Anything 2 (Ours)** | Same as FTP-VM | 18.01 | 4.89 | 26.26 | 2.76 | 1.08 | 0.22 | 2.64 | 1.20 |
| **Matting Anything 2 (Ours)** | Respective Sets | 14.72 | 3.70 | 23.54 | 2.65 | 1.16 | 0.24 | 3.07 | 1.20 |

## A.5 LOSS FUNCTION

The training of MAM2 involves three loss functions: $L_{\mathrm{mask}}$, $L_{\mathrm{trimap}}$, and $L_{alpha}$.

Among these, $L_{\mathrm{mask}}$ is inherited from the original SAM2, which is a combination of a focal and dice loss for the mask, a MAE loss for the predicted IoU score, and a cross-entropy loss for object prediction.

For $L_{\mathrm{trimap}}$, we adopt the Normalized Focal Loss, a loss function commonly used in interactive segmentation Sofiiuk et al. (2022); Liu et al. (2023), which is defined as follows:

$$\mathrm{NFL}(i,j) = -\frac{1}{\sum_{i,j}(1-p_{i,j})^\gamma}(1-p_{i,j})^\gamma \log p_{i,j} \qquad (8)$$

where $p_{i,j}$ denotes the confidence at $(i,j)$ of the predicted $trimap \in \mathbb{R}^{W \times H \times 3}$.

$L_{alpha}$ loss is a combination of separate $l1$ loss Yao et al. (2024), $l2$ loss, Laplacian loss Hou & Liu (2019), and gradient penalty loss Tang et al. (2019), formulated as follows

$$L_{alpha} = L_{separate\ l1} + L_{l2} + L_{laplacian} + L_{gradient}.$$

## B NATURAL VIDEO MATTING DATASET

Table 9: Breakdown of the NOVM dataset.

| Category | Animals | Bubble | Cloud | Explosion | Fire | Frost | Plant | Slime | Vehicles | Water | Sum |
|---|---|---|---|---|---|---|---|---|---|---|---|
| Quantity | 9 | 4 | 4 | 5 | 4 | 4 | 7 | 2 | 4 | 7 | 50 |
| Proportions | 18% | 8% | 8% | 10% | 8% | 8% | 14% | 4% | 8% | 14% | 100% |

The most distinctive feature of our proposed NOVM dataset is its category diversity, as illustrated in Table 9. Furthermore, we have balanced the number of samples per category to ensure a uniform

class distribution. We provide several examples from the NOVM dataset in Fig. 7, which demonstrate that the domain diversity of its objects far surpasses that of existing video matting benchmarks, and that its alpha mattes are of exceptional quality.

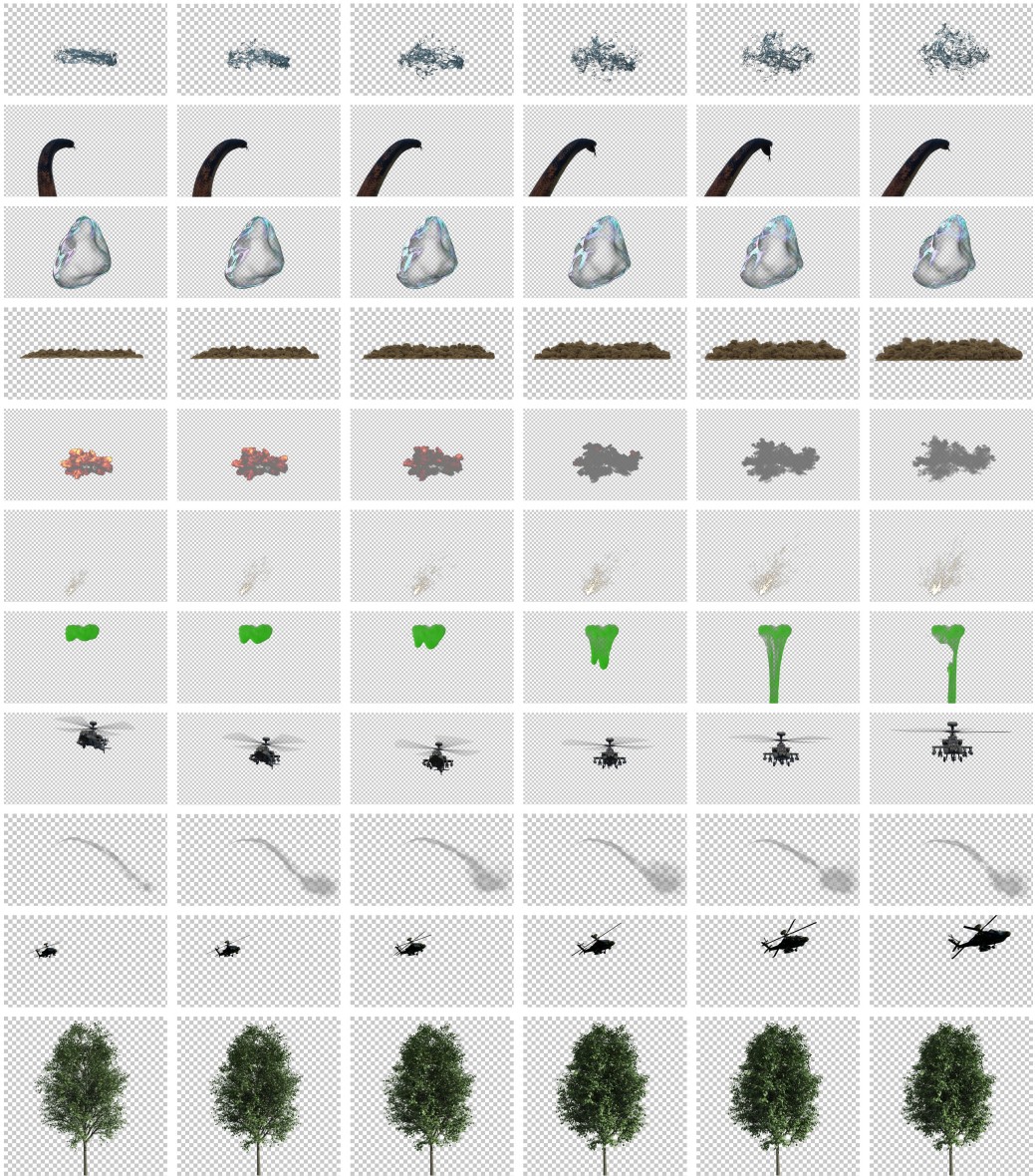

Figure 7: Examples of NOVM dataset.

## C VISUALIZATION

### C.1 VIDEO MATTING

We present the visualization results of MAM2 on the video matting task in Fig. 8 and 9, displaying both the predicted alpha mattes and the extracted foregrounds. Additionally, we provide visualization results on the YouTubeMatte dataset in Fig. 10, and on real-world videos in Fig. 11 and 12. We provide visual comparison with other methods on YoutubeMatte in Fig. 13. Given the limited

fine details in public matting benchmarks, we have supplemented the last row of Fig. 13 with the video characterized by large-scale motion and complex hair structures. Several visualization results in MP4 format are also provided in the supplementary material.

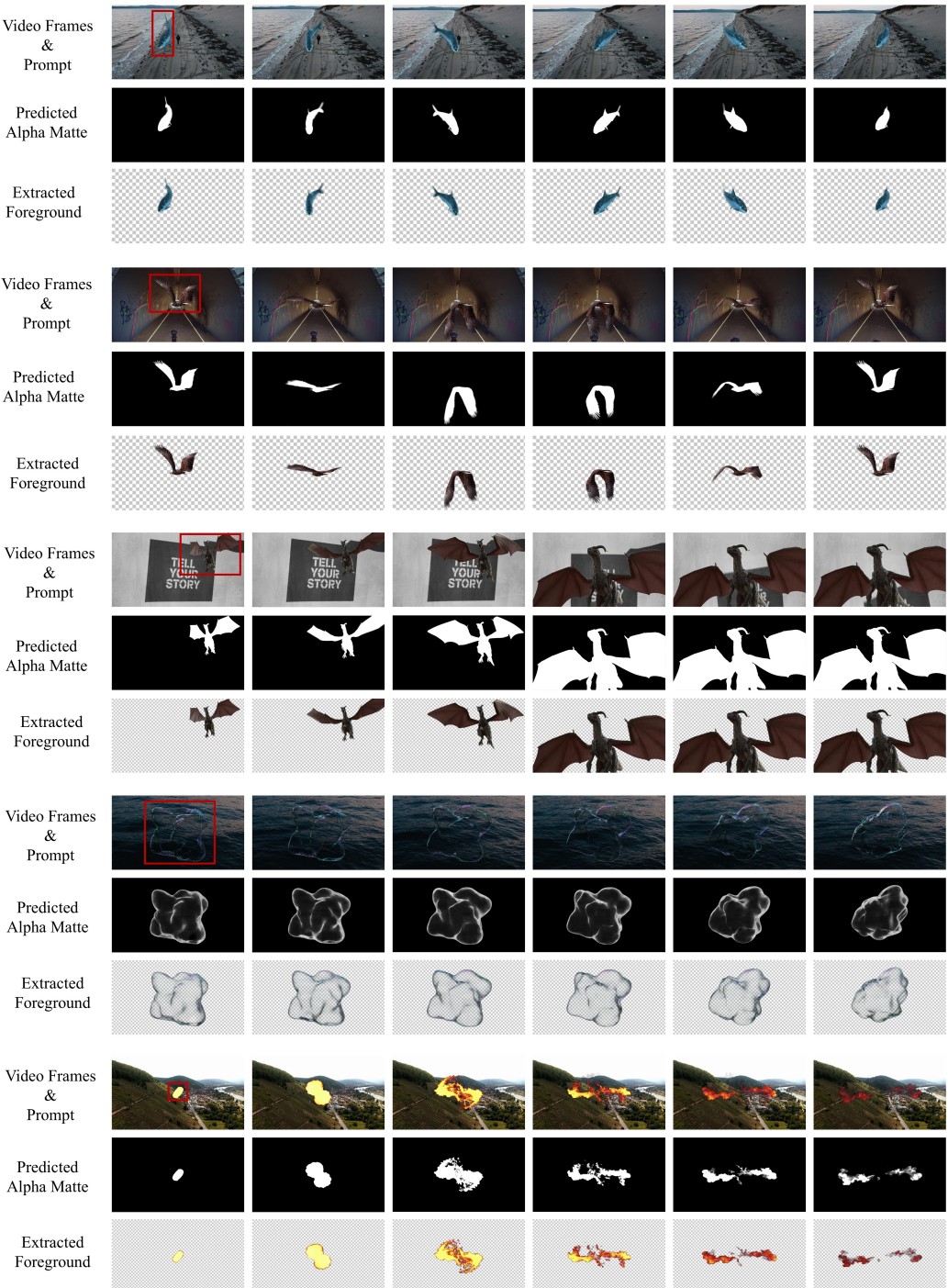

Figure 8: Visualization results of MAM2 on NOVM.

## C.2 IMAGE MATTING

We present MAM2's visualization results of MAM2 on the image matting task in Fig. 14,

## C.3 VIDEO EDITING

Since video matting empowers creators with control at the layer and object levels, it is extensively applied in various creative editing workflows. In Fig. 15, we present several simple video editing effects realized using MAM2, including background replacement, trajectory control and object scaling. These examples represent merely the tip of the iceberg regarding matting-based editing; experienced and creative practitioners can undoubtedly leverage matting to generate far more sophisticated and visually stunning effects.

## D LATENCY

We compared the FPS (Frames Per Second) and parameter count of MAM2 against recent methods in Table 10, with all tests conducted on an NVIDIA RTX 5880 GPU. While MAM2 is not as computationally efficient, it achieves significantly higher matting accuracy. In practical deployment, methods requiring a first-frame mask necessitate the deployment of auxiliary models to generate this mask. For instance, the official MatAnyone demo deploys an additional SAM-ViT-Huge Kirillov et al. (2023), while MaGGIe utilizes SAM 2-Base Ravi et al. (2024). Therefore, from the perspective of practical deployment, we have included these auxiliary parameters to compare the total parameter count required by each method.

| | MAD ↓ | Grad ↓ | FPS ↑ | Params ↓ | Params of Auxiliary Model ↓ | Total Params ↓ |
|---|---|---|---|---|---|---|
| MaGGIe | 50.04 | 108.01 | **18.87** | **30.91** | 80.80 | **111.71** |
| MatAnyone | 39.44 | 25.63 | 15.43 | 35.25 | 636.00 | 671.25 |
| Matting Anything 2 | **14.72** | **3.70** | 11.24 | 256.83 | **0** | 256.83 |

Table 10: Efficiency comparison with other methods.

## E LIMITATIONS

Despite MAM2 achieving promising performance, as a pioneering attempt at video matting across such diverse domains, it inevitably exhibits certain limitations. These issues primarily manifest when the target object possesses high transparency. For instance, in the explosion shown in the left part of Fig. 16, as the smoke dissipates, its opacity becomes extremely low. This results in a weak visual signal, causing the model to abruptly fail in capturing a significant portion of the smoke. As illustrated in the third column, while smoke actually persists with high transparency in the upper section of the ring, it is missing from the extracted result.

Another issue pertains to the extraction of transparent objects based on the alpha matte. Since the standard practice in matting involves extracting the foreground by multiplying the original image with the predicted alpha matte, the resulting foreground inevitably retains some background color. For example, in the right part of Fig. 16, the shape of the background mountains remains faintly visible along the edges of the extracted bubble, albeit slightly, and becomes even less discernible when composited onto a new background. We consider this an open problem in the field of matting worth future exploration: specifically, how to extract a clean foreground based on the alpha matte.

## F LLM USAGE

LLMs are used for refining English usage, and all content is reviewed by authors.

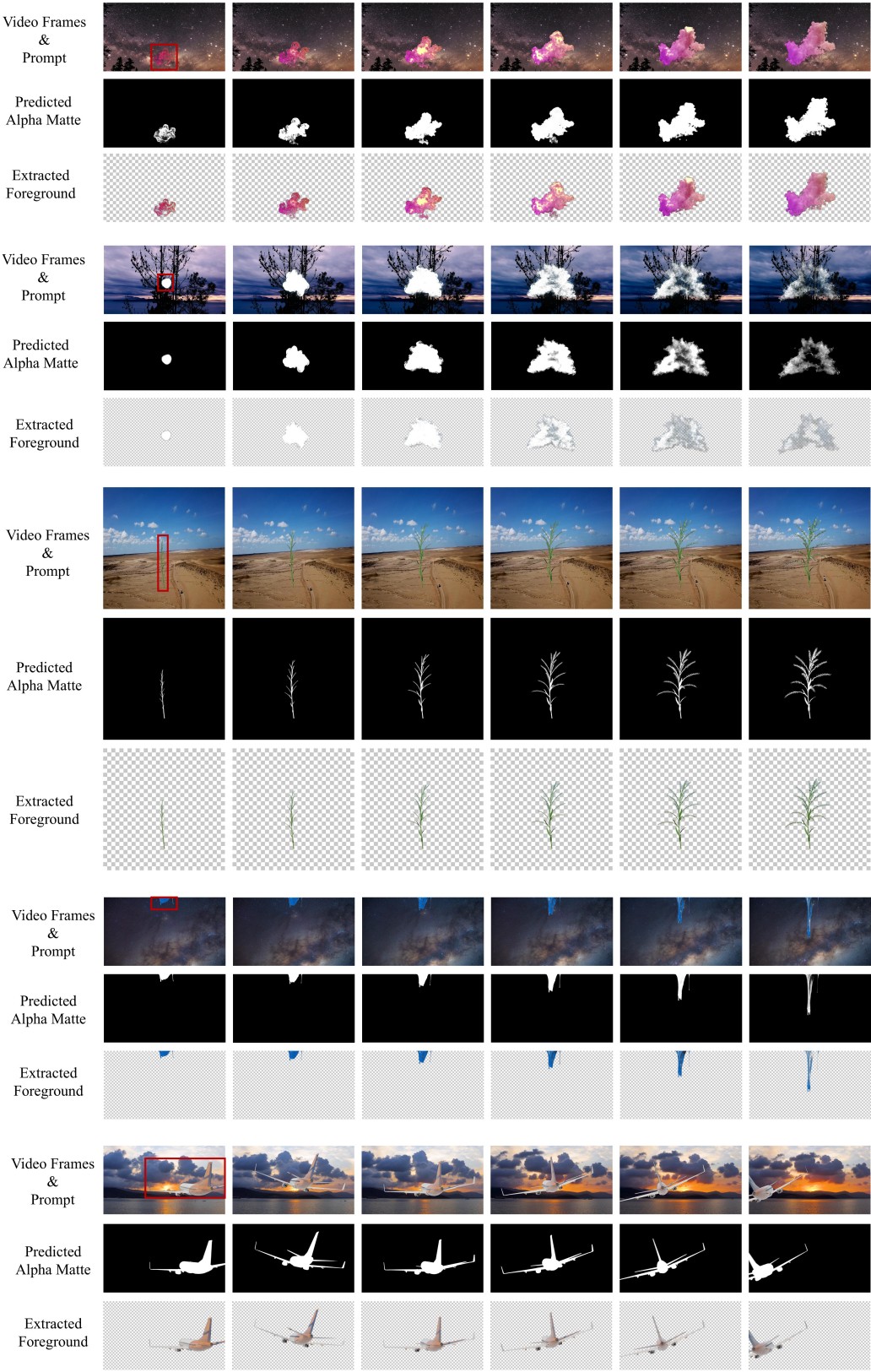

Figure 9: Visualization results of MAM2 on NOVM.

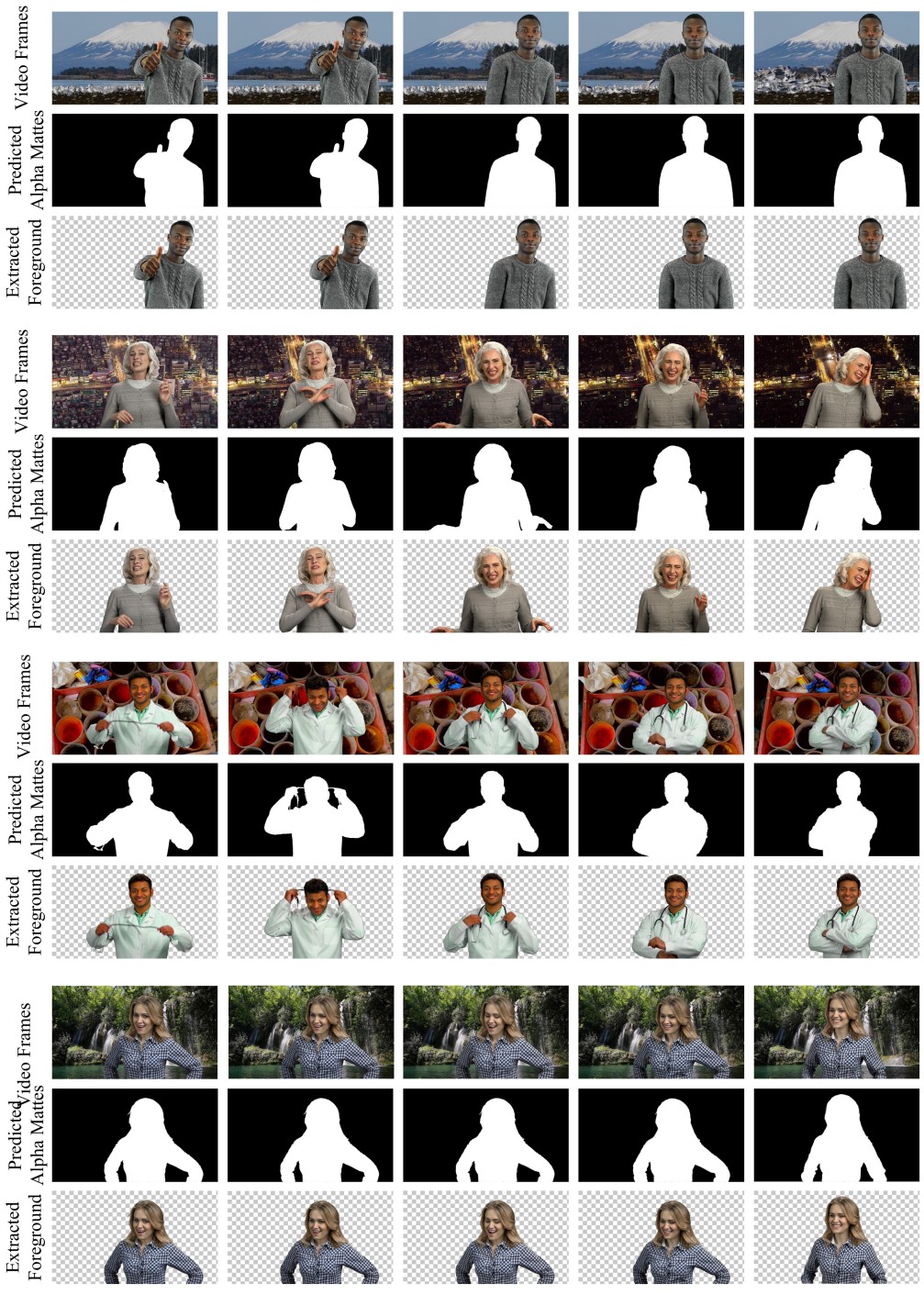

Figure 10: Visualization results of MAM2 on YoutubeMatte.

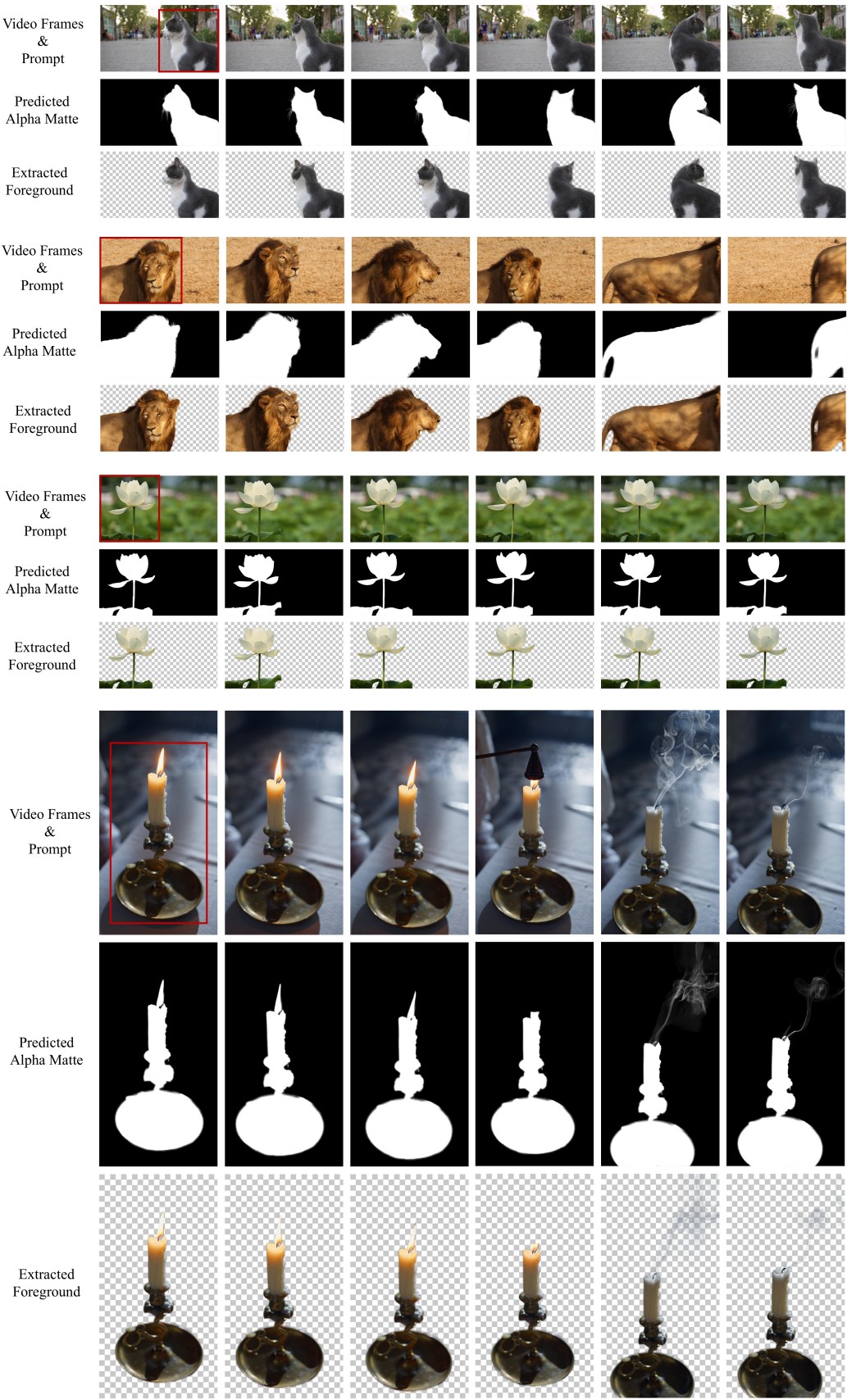

Figure 11: Visualization results of MAM2 on real-world videos.

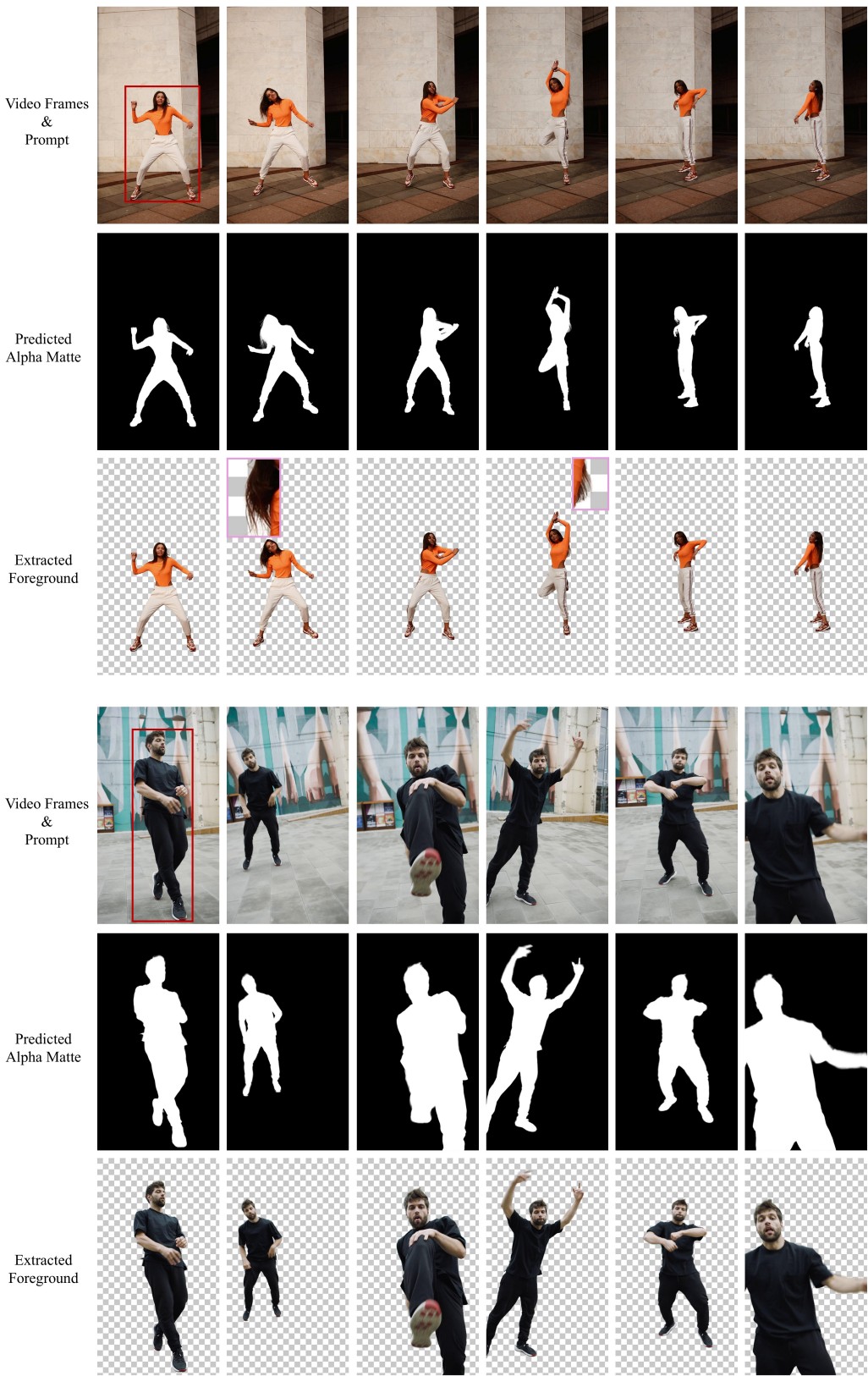

Figure 12: Visualization results of MAM2 on real-world videos.

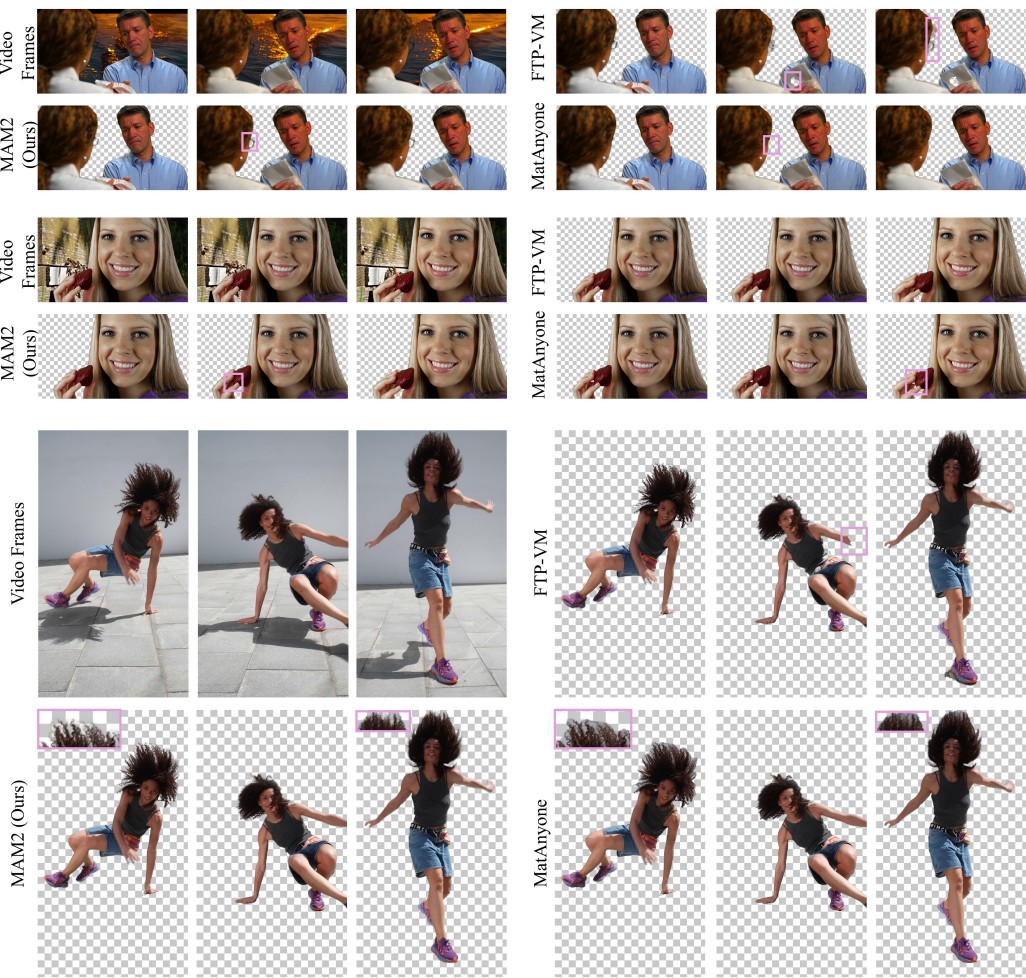

Figure 13: Visualization comparison with other methods.

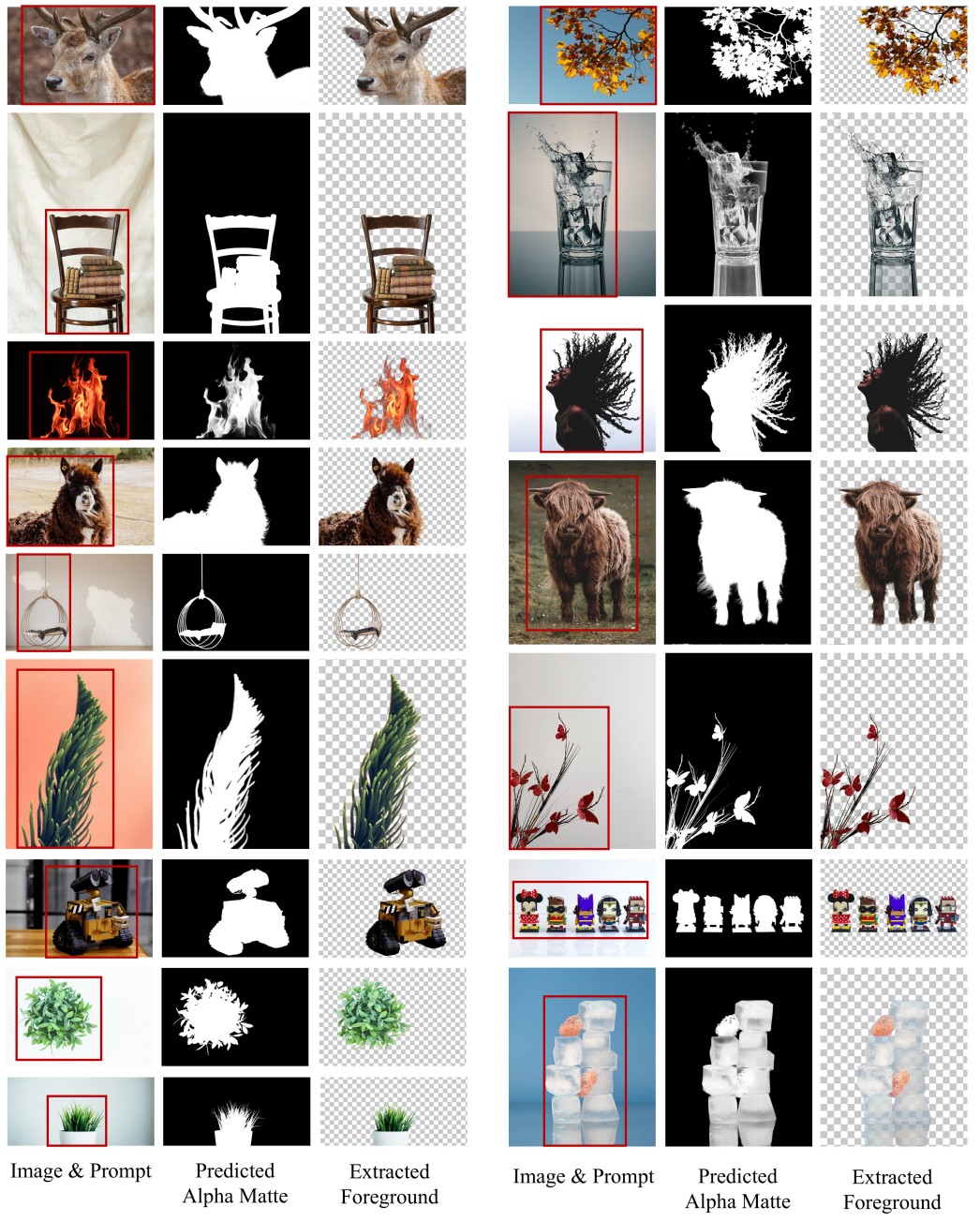

Image & Prompt    Predicted     Extracted       Image & Prompt    Predicted     Extracted
                  Alpha Matte   Foreground                        Alpha Matte   Foreground

Figure 14: Visualization results of MAM2 on image matting.

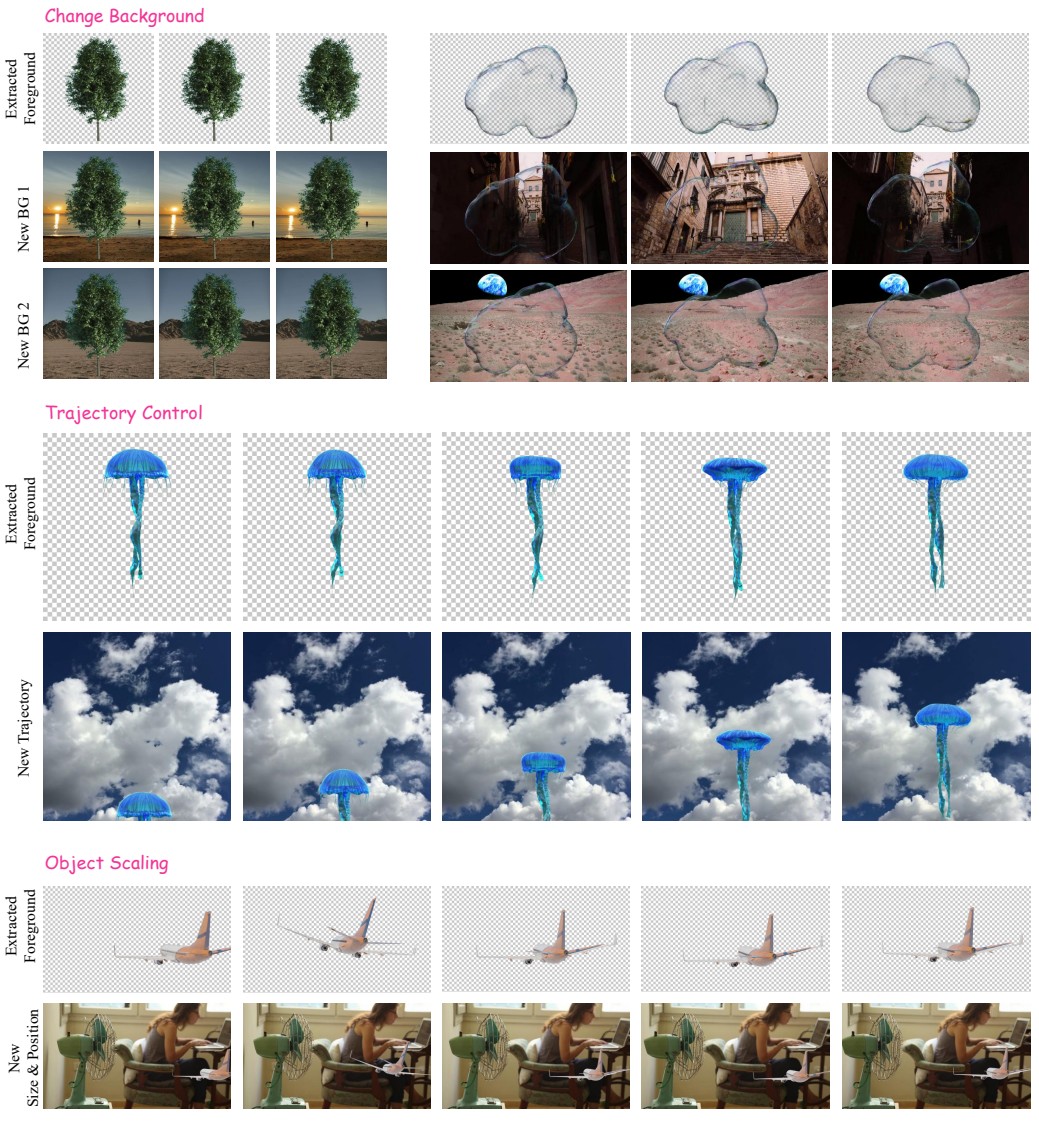

Figure 15: Simple matting-based video editing effects.

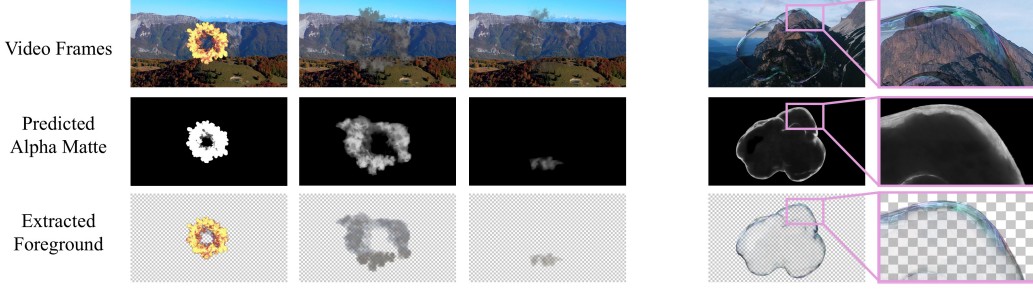

Figure 16: Visualization results of limitations.

