# OpenReview forum: "Matting Anything 2:  Towards Video Matting for Anything"
_ICLR.cc/2026/Conference — ICLR 2026 Poster_

### Official Review · Reviewer_TLqf · 2025-10-27

**Soundness:** 2
**Presentation:** 1
**Contribution:** 2
**Rating:** 4
**Confidence:** 4

**Summary:**

This work introduces Matting Anything 2 known as MAM2, a prompt driven video matting framework that moves beyond human portrait centric settings by supporting points, boxes, and masks, by stabilizing trimap prediction for transparent and complex objects, and by reducing reliance on a first frame mask. The method adds a Promptable Dual-mode Decoder that predicts a segmentation mask and a trimap in one pass, and a Memory Separable Siamese mechanism that stabilizes trimap decoding for transparent or complex objects across time. The authors also introduce a Natural Video Matting benchmark with diverse non human portrait categories. Results report strong accuracy on both natural objects and human videos, and competitive image matting as well.

**Strengths:**

1. Clear problem motivation regarding portrait bias and reliance on a first frame mask for selection. The paper explains why transparent targets such as smoke or fire make mask prompting difficult and why lighter prompts like box or points are preferable for such cases.

2. Architectural idea with practical value. PDD extends the SAM2 mask decoder to produce both a mask and a trimap in one pass, and it leverages the strong mask quality of SAM2 as a spatial prior. In practice this yields cleaner boundaries and a more stable unknown band, which improves the final alpha matte.

3. Transparent object failure analysis and remedy. The paper identifies temporal collapse where unknown regions drift to foreground for later frames. MSS addresses this by running a second PDD pass to decode the trimap from memory free features using the first pass mask as a pseudo prompt. Parameters are shared between the passes.

4. New test dataset (Natural Video Matting) for generalization.

**Weaknesses:**

1. The mathematical specification of the pipeline is insufficient.
The paper lacks a complete equation level description of the forward computations, especially for the interaction between PDD and MSS.

2. Figures do not explain the full system behavior. Figure 2 leaves ambiguity about how segmentation data and matting data are used across training iterations and stages. It is unclear how features from the MSS pathway connect to the PDD pathway, which parts are trainable at each stage, and whether user prompts for MSS and PDD are shared or distinct. The figure should be redrawn to show the end to end pipeline with data sources, feature flow, prompt flow, and trainable versus frozen modules.

3. The boundary between the SAM2 decoder and the authors’ contributions is unclear.
The text and figures do not make it evident what is inherited from the original SAM2 mask decoder and what is newly introduced in this work. The paper should provide a module level figure that clearly labels inherited blocks and newly added blocks, together with explicit annotation of the mask output token, the trimap output token, and all entry points for user prompts.

4. Why the method works is not analyzed.
The paper states that the proposed method fixes the failure cases but gives little analysis of the mechanism. In particular, the paper should explain why the Mask Augment Feature and the preserved Feature without Memory lead to stable trimap prediction, with diagnostic evidence or ablations.

5. Experimental fairness and data accounting are not sufficiently documented.
According to Appendix 1, models appear to be trained on different data sources. The manuscript should quantify the training data for each method in comparable units such as number of images, clips, and frames, and include experiments where all methods are trained on the same data to isolate the effect of the proposed design. Table 8 also indicates a parameter gap of nearly nine times between MAM2 and the strongest baseline. This capacity difference makes it difficult to attribute the gains in MAD and GRAD to the proposed architecture rather than to model size.

**Questions:**

1. Figure 5 appears to report results only on the Natural Video Matting dataset. Could you add a companion figure that compares MAM2 with baselines on additional public video matting test sets?

2. Table 2 shows MAM2 performance with box and point prompts. In Table 4 these entries appear in faint text, which suggests they may be out of scope or not directly comparable. Could you create a separate table summarizing results for models that use both prompts box and point under the same data and evaluation protocol, reporting MAD and GRAD, so the two prompt setting can be compared fairly with single prompt settings?

3. Could you revise Figure 4 to also display the matted image together with the trimap for the same frames and prompts so readers can directly see how trimap quality translates into the visual result?

4. There are some typographical errors. For example, Figure 7,8,9 caption contains “visualizaiotn” instead of “visualization”. Could you proofread the paper and update all figure texts and captions, ensuring correct spelling, consistent capitalization?

5. Section 3.4 is difficult to read and follow. Could you provide a clearer rewrite during rebuttal?

---

> ### Author Response · Authors · 2025-11-23
> **Official Comment by Authors (Part 1)**
>
> We sincerely appreciate the time you spent reviewing our paper and providing  valuable feedback. We have carefully responded to each of your comments and updated the manuscript. We hope that these revisions satisfactorily address your concerns.
>
>
> >**W1: Mathematical specification of the pipeline is insufficient**
>
> We appreciate this constructive criticism. In the previous version of the manuscript, mathematical descriptions were limited to the training strategy and loss functions, while the model architecture was described purely in text. We realize this lack of formalism was likely a primary cause of the confusion regarding the PDD and MSS modules. Consequently, we have substantially revised the sections corresponding to PDD (Section 3.4) and MSS (Section 3.5), and added a new Preliminary section (Section 3.3) to facilitate the explanation. However, to ensure the readability of the PDF file, we have not highlighted the entire sections in blue. We have incorporated comprehensive mathematical formulas throughout these three sections to rigorously clarify the mechanisms.
>
> >**W2.1: How segmentation data and matting data are used across training iterations and stages?**
>
> We have added a new section in Appendix A.2 to detail how different datasets are utilized across various training stages, accompanied by a new Figure 6 for illustration.
>
> >**W2.2:  How features from the MSS pathway connect to the PDD pathway?**
>
> We believe this misunderstanding constitutes the primary source of confusion. To address this, we have substantially revised the sections on PDD and MSS to provide a detailed explanation. Here, we offer a concise clarification:
>
> There is **no feature connection** between PDD and MSS because MSS is not a distinct module or pathway; rather, it is a parameter-free strategy. The essence of MSS lies in invoking the PDD twice in sequence.
>
> Specifically, for the decoding of subsequent frames, we first invoke the PDD using memory-embedded features to obtain preliminary results. Subsequently, we utilize the just-predicted mask to drive a second PDD pass. Crucially, the input provided to the PDD for this second pass consists of *memory-free* features. This distinction is vital because the original SAM 2 mechanism—which embeds memory from previous predictions into current features—induces a significant shift in the feature space. Consequently, trimaps predicted based on such shifted features tend to resemble binary masks, frequently resulting in the erroneous classification of *unknown* regions as *foreground*.
>
> MSS can be formally described by the following equation:
>
> $$
> \begin{array}{rl}
> M^t & = \pi_1\left( f _{\text{PDD}}(\mathbf{F} _{\text{mem}}^t, \emptyset) \right) \\\\
> T^t & = \pi_2\left( f _{\text{PDD}}(\mathbf{F} _{\text{non-mem}}^t, M^t) \right)
> \end{array}
> $$
>
> where $M^t$ and $T^t$ denote the predicted mask and trimap at frame $t$, respectively. $\mathbf{F} _{\text{mem}}^t$ and $\mathbf{F} _{\text{non-mem}}^t$ represent the image features **with** and **without** memory embedding. $\pi_1$ and $\pi_2$ denote the projection functions that extract the first (Mask) and second (Trimap) outputs from the PDD.
>
> This structural design is also clearly illustrated in part (c) of the revised Figure 2. This design rationale aligns precisely with the 'Memory-Separable' designation. Crucially, while PDD represents the architectural module, MSS represents the strategy for PDD's invocation.
>
> >**W2.3: Which parts are trainable at each stage**
>
> Specifically, $\Theta_{\text{main}}$, which is responsible for trimap prediction, is trainable in Stage 1, while $\Theta_{\text{matter}}$, used for alpha matte prediction, is trainable in Stage 2. We have also annotated this distinction in Figure 6.
>
>  >**W2.4: Whether user prompts for MSS and PDD are shared or distinct**
>
> As discussed in W2.2, MSS is essentially an invocation strategy for the PDD and MSS itself does not inherently accept user prompts. We acknowledge that the previous version of Figure 2, which depicted user prompts as inputs for both PDD and MSS components, likely caused this confusion. In the revised version, we have placed the user prompt input exclusively in Part (a) of Figure 2, ensuring it appears only once to avoid ambiguity. Furthermore, the newly added mathematical formulas also facilitate the reader's understanding.
>
>  >**W3: The boundary between the SAM2 decoder and the authors’ contributions is unclear**
>
> In Part (b) of the revised Figure 2, the dashed lines represent the mask decoding flow inherited from the SAM 2 decoder, while the solid lines illustrate MAM2's specific trimap decoding flow that operates beyond the SAM 2 decoder. Furthermore, the MSS mechanism depicted in Part (c) represents a completely novel design that is not inherited from SAM 2.

---

> > ### Author Response · Authors · 2025-11-23
> > **Official Comment by Authors (Part 2)**
> >
> > >**W4: Why the method works**
> >
> > We have provided a clear motivation and detailed analysis in Sections 3.4 and 3.5 of the revised manuscript. Furthermore, we provide comprehensive ablation studies in Table 5. Here, we offer a concise summary:
> >
> > Why does PDD work: We observed that simply adding a parallel trimap decoding branch to the SAM 2 mask decoder results in noisy trimap predictions. However, the masks predicted by the mask branch remain consistently stable. To transfer this stability to the trimap prediction, we transform the predicted mask into a mask augment feature via a simple convolution and inject it into the trimap features. This strategy effectively enhances the stability of trimap decoding.
> >
> > Why does MSS work: We observed that while prediction quality for transparent objects is typically high in the first frame, it suffers from collapse starting from the second frame. Consequently, we analyzed the original temporal decoding mechanism of SAM 2. As discussed in Section 3.3, the fundamental difference lies in the driving prompt: the first frame is driven by user prompts, whereas subsequent frames are driven by memory embedded within the features. Therefore, we hypothesized that this memory embedding interferes with trimap decoding. Based on this, we attempted to decode subsequent frames using memory-free features. We introduced MSS to solve the challenge of driving the PDD without memory embedding. In essence, MSS strategically avoids memory interference during trimap decoding while ensuring continuous temporal propagation.
> >
> >
> >
> >
> >  >**W5.1: Experimental fairness and data accounting**
> >
> > In Table 6, we provide the composition of the training data used by each baseline method, along with the total number of images and videos utilized, as shown in the table below.
> >
> > | Methods | Total Images | Total Videos | YoutubeVIS | MOSE | VM800 | VM108 | VM240K | CRGNN | COCO | SPD | DUTS | DIS5K | DIM | D-646 | AM-2K | T-460 | I-HIM50K | P10K |
> > | :--- | :---: | :---: | :---: | :---: | :---: | :---: | :---: | :---: | :---: | :---: | :---: | :---: | :---: | :---: | :---: | :---: | :---: | :---: |
> > | **FTP-VM** | 596 | 3551 | ✓ | | | ✓ | | | | | | | | ✓ | | | | |
> > | **MAGGIE** | 49373 | 575 | | | | ✓ | ✓ | ✓ | | | | | | | | | ✓ | |
> > | **MatAnyone**| 89251 | 4297 | ✓ | | ✓ | | | | ✓ | ✓ | | | ✓ | ✓ | | | | |
> > | **MAM2 (Ours)** | 12239 | 1326 | | ✓ | | ✓ | | | | | ✓ | ✓ | ✓ | ✓ | ✓ | ✓ | | ✓ |
> >
> > As shown in the table above, all baseline methods employ distinct training dataset compositions. Given that some datasets are private and not publicly available (e.g., the VM800 dataset used by MatAnyone), and that training codes for certain methods are unreleased, it is infeasible to standardize the training settings across all methods for comparison.
> >
> > However, to demonstrate that our performance improvements are not merely artifacts of inconsistent training data, we aligned our training setting with that of FTP-VM—which utilizes the most straightforward data composition (consisting solely of VM108, YouTubeVIS, and D-646)—and retrained MAM2. The results are presented below. As observed, our method maintains significantly superior performance, confirming that MAM2's gains stem from its design rather than data advantages. We have incorporated this comparison into Table 8 of the revised manuscript.
> >
> > | Method | Training Data | NOVM MAD ↓ | NOVM MSE ↓ | NOVM GRAD ↓ | NOVM dtSSD ↓ | Youtube MAD ↓ | Youtube MSE ↓ | Youtube GRAD ↓ | Youtube dtSSD ↓ |
> > | :--- | :--- | ---: | ---: | ---: | ---: | ---: | ---: | ---: | ---: |
> > | TCVOM | Respective Sets | 56.18 | 38.90 | 153.95 | 3.84 | 1.57 | 0.40 | 6.74 | 1.52 |
> > | FTP-VM | Respective Sets | 37.98 | 19.90 | 78.06 | 4.24 | 2.26 | 1.10 | 5.63 | 1.70 |
> > | MAGGIE | Respective Sets | 50.04 | 35.23 | 108.01 | 4.90 | 2.37 | 0.98 | 7.69 | 1.77 |
> > | MatAnyone | Respective Sets | 39.44 | 25.63 | 89.60 | 4.10 | 2.05 | 0.76 | 9.67 | 1.75 |
> > | **Matting Anything 2 (Ours)** | Same as FTP-VM | 18.01 | 4.89 | 26.26 | 2.76 | 1.08 | 0.22 | 2.64 | 1.20 |
> > | **Matting Anything 2 (Ours)** | Respective Sets | 14.72 | 3.70 | 23.54 | 2.65 | 1.16 | 0.24 | 3.07 | 1.20 |

---

> > > ### Author Response · Authors · 2025-11-23
> > > **Official Comment by Authors (Part 3)**
> > >
> > > >**W5.2:  Gap in parameter count**
> > >
> > > A core advantage of MAM2 is its ability to perform video matting directly based on user prompts (e.g., boxes and points) without relying on any auxiliary models. In contrast, baseline methods typically require an initial mask or trimap for the first frame, necessitating auxiliary segmentation models to generate these inputs. For instance, the official demo of MatAnyone [1] deploys an additional SAM-ViT-Huge as an auxiliary model, while MaGGIe [2] utilizes both SAM 2-Base and XMem  [3] as auxiliary models.
> > >
> > > Therefore, from the perspective of practical deployment, we have calculated the total parameter counts required by the baseline methods, as shown in the table below. It is evident that MAM2's seemingly higher parameter count arises because it integrates prompt understanding internally. When considering the total deployment cost, MAM2 does not exhibit a significant disadvantage in parameter count.
> > >
> > > | Method | MAD ↓ | Grad ↓ | FPS ↑ | Params ↓ | Params of Auxiliary Model ↓ | Total Params ↓ |
> > > | :--- | :---: | :---: | :---: | :---: | :---: | :---: |
> > > | MAGGIE | 50.04 | 108.01 | **18.87** | **30.91** | 59.37 + 80.80 | **171.08** |
> > > | MatAnyone | 39.44 | 25.63 | 15.43 | 35.25 | 636.00 | 671.25 |
> > > | Matting Anything 2 | **14.72** | **3.70** | 11.24 | 256.83 | **0** | 256.83 |
> > >
> > >
> > > [1] MatAnyone: Stable Video Matting with Consistent Memory Propagation, CVPR 2025
> > >
> > > [2] MaGGIe: Mask Guided Gradual Human Instance Matting, CVPR 2024
> > >
> > > [3] XMem: Long-Term Video Object Segmentation with an Atkinson-Shiffrin Memory Model , ECCV 2022
> > >
> > >
> > >
> > >
> > >
> > > >**Q1: Visual comparison on other public test sets**
> > >
> > > We have added visual comparisons with other methods on the YoutubeMatte test set in Figure 13.
> > >
> > > >**Q2: Prompt Setting**
> > >
> > > To clarify: Table 2 is for Video Matting and Table 4 is for Image Matting. These two tables represent separate evaluations with no connection.. Additionally, SDMatte (in faint text) is a baseline method, not ours. It appears 'Box & Text' might have been misread as 'Box & Points'.
> > >
> > > We strongly agree that comparisons must use consistent prompt settings. We displayed the 'Box & Text' results for SDMatte in faint text because SDMatte is the only method that requires and is capable of accepting text prompts. Apart from this exception, all comparisons in Table 4 are conducted using identical prompt settings. This principle also applies to Table 2: while MAM2 can operate solely on Boxes and Points, we also reported its Mask-based results specifically to ensure a fair comparison with other mask-dependent methods.
> > >
> > > >**Q3: Display the matted image in Figure 4**
> > >
> > > We agree with your reasonable suggestion. Accordingly, we have updated Figure 4 in the revised manuscript to include the matted images.
> > >
> > > >**Q4: Typographical errors**
> > >
> > > We sincerely apologize for the typographical errors that were inadvertently left in the manuscript. We have thoroughly proofread the full text and corrected these errors in the revised version.
> > >
> > > >**Q5: Section 3.4 is difficult to read and follow**
> > >
> > > The difficulty in reading the MSS section likely arose from the previous layout, which mixed the background analysis of SAM 2 with the description of MSS. To fix this, we adjusted the structure in the revised manuscript. We added a standalone Preliminary section (Section 3.3) to handle the analysis of SAM 2's temporal decoding mechanism first. This allows the MSS section (Section 3.5) to focus exclusively and directly on the proposed mechanism. We hope this clearer structure aids understanding.

---

> ### Author Response · Authors · 2025-11-24
> **Official Comment by Authors**
>
> Thank you for noticing this detail. The misalignment was indeed caused by a clerical error where we inserted the wrong image. We sincerely apologize and have fixed this in the revised manuscript.
>
> This is purely a display error and does not represent any specific phenomenon. Your interpretation is correct.

---

> > ### Comment · Reviewer_TLqf · 2025-11-25
> >
> > Authors have provided clear explanations and additional evidence that resolve the issues I previously raised. Based on these improvements, I am raising my score.

---

### Official Review · Reviewer_Uqg2 · 2025-10-31

**Soundness:** 3
**Presentation:** 2
**Contribution:** 2
**Rating:** 2
**Confidence:** 4

**Summary:**

This paper proposes a generic video matting algorithm for different objects including person, animal, fire, water etc. It proposes a two-branch network to conduct the segmentation and tri-map prediction tasks, respectively. It also proposes a new dataset named Natural Video Matting covering different object categories.

The generic video matting task is challenging, and the authors take an initial step towards it.

**Strengths:**

-	This work deals with general objects for video matting, which is more advanced than existing work mainly dealing with humans.
-	According to table 2, the proposed method (matting anything2) outperforms existing methods on the proposed new dataset and a human matting dataset.

**Weaknesses:**

-	The model architecture is largely Segment Anything 2 (SAM2) with some additional components. The box and point prompt capability are directly from SAM2. The novelty of the proposed framework is limited.
-	The evaluation dataset is small for general object categories. It only contains 50 clips.
-	In supplementary section A, it shows the proposed method and existing method uses different datasets for training the model. The proposed method used more image matting data for training, which may pose an unfair comparison with existing methods. Can authors provide a variant that is trained with same data (like what Matanyone used) to give a fair comparison?

**Questions:**

see above

---

> ### Author Response · Authors · 2025-11-23
> **Official Comment by Authors (Part 1)**
>
> We appreciate the comments regarding our work. We have responded to your comments point-by-point and revised the manuscript accordingly. We hope that these revisions address your concerns.
>
>
> > **W1: The novelty of the proposed framework is limited.**
>
> We respond to this concern from two perspectives:
>
> **From the perspective of overall architecture:**
>
> MAM2 utilizes SAM 2 as a foundation specifically to leverage its robust semantic understanding and superior user interactivity. Beyond this, the overall architectural philosophy of MAM2 diverges significantly from SAM 2. The most distinct difference lies in the prediction pipeline: Although SAM 2 is capable of predicting high-quality **masks**, MAM2 deliberately does not choose to predict the **alpha mattes** based on the **masks**. Instead, it first predicts **trimaps** and subsequently predicts the **alpha mattes**. This approach distinguishes MAM2 not only from SAM 2 but also from recent video matting methods [1,2], which predominantly rely on mask-based alpha prediction. We posit that a standard adaptation of SAM 2 for video matting would likely follow this prevailing mask-based trend, rather than adopting the unique trimap-based architecture we propose in MAM2.
>
> Furthermore, while earlier methods [3,4,5] have explored trimap-based video matting, they typically require users to manually draw the trimap for the first frame. In contrast, MAM2 inherits the efficient interactive capabilities of SAM 2, allowing users to simply provide a box or points to automatically generate the trimap. This significantly enhances interaction efficiency.
> In summary, while MAM2 inherits beneficial properties from SAM 2, its core pipeline—**User Prompt $\rightarrow$ Trimap $\rightarrow$ Alpha Matte**—represents a pioneering approach.
>
> **From the perspective of specific structural components:**
>
> In terms of implementation, adapting SAM 2 to predict Trimaps is non-trivial. To the best of our knowledge, predicting trimaps based on SAM has only been explored by SEMatte [6], which is an *image matting* model (not *video*), and it does not yield high-quality trimap predictions, primarily because the trimap serves only as an auxiliary output in SEMatte. To address the challenges in the video domain, we proposed the **PDD** module and **MSS** mechanism to specifically enhance the stability and temporal consistency of trimap prediction. These mechanisms are novel contributions to the matting field, and we believe they offer valuable insights to the community.
>
> Therefore, considering both the novel overall architectural strategy and the effective specific modules, we believe MAM2 demonstrates sufficient novelty.
>
>
>
> >**W2: evaluation dataset is small for general object categories**
>
> We acknowledge that compared to large-scale segmentation datasets, 50 clips may appear limited. However, within the filed of Video Matting, a test set of 50 clips is relatively substantial. As demonstrated in the table below, our dataset actually exceeds the size of other popular video matting benchmarks.
>
> | Test Set | Count | Average Duration | Domain |
> | :--- | :---: | :---: | :--- |
> | VideoMatte240K | 25 | 100 | human |
> | VideoMatting108 | 28 | 845 | human, cloth, smoke |
> | YoutubeMatte | 32 | 100 | human |
> | **Natural Object Video Matting (Ours)** | **50** | 164 | animals, bubble, cloud, fire, water, frost, vehicles, plant... |
>
> Furthermore, we have intentionally maximized category diversity while ensuring a balanced distribution of samples across categories, as shown in the Table below. Notably, we deliberately excluded human subjects, as this category is already predominantly represented in existing datasets. We have made every effort to provide a challenging benchmark that covers domains largely absent in current literature.
>
> | Category | Animals | Bubble | Cloud | Explosion | Fire | Frost | Plant | Slime | Vehicles | Water | Sum |
> | :--- | :---: | :---: | :---: | :---: | :---: | :---: | :---: | :---: | :---: | :---: | :---: |
> | Quantity | 9 | 4 | 4 | 5 | 4 | 4 | 7 | 2 | 4 | 7 | 50 |
> | Proportions | 18% | 8% | 8% | 10% | 8% | 8% | 14% | 4% | 8% | 14% | 100% |
>
> [1] MatAnyone: Stable Video Matting with Consistent Memory Propagation, CVPR 2025
>
> [2] MaGGIe: Mask Guided Gradual Human Instance Matting, CVPR 2024
>
> [3] End-to-End Video Matting With Trimap Propagation, CVPR 2023
>
> [4] One-Trimap Video Matting, ECCV 2022
>
> [5] Attention-guided Temporally Coherent Video Object Matting, ACMMM 2021
>
> [6] Towards Natural Image Matting in the Wild via Real-Scenario Prior, arxiv

---

> > ### Author Response · Authors · 2025-11-23
> > **Official Comment by Authors (Part 2)**
> >
> > >**W3: provide a variant that is trained with same data**
> >
> > Your suggestion is well-founded; variations in training data indeed impact the results. However, since recent methods utilize highly divergent training sets, it is infeasible to unify all methods under a single training setting. Nevertheless, following your suggestion, we aligned our training set with that of FTP-VM[3]. As shown in Table below, our method still achieves significantly superior performance under this aligned setting.
> >
> > | Method | Training Data | NOVM MAD ↓ | NOVM MSE ↓ | NOVM GRAD ↓ | NOVM dtSSD ↓ | Youtube MAD ↓ | Youtube MSE ↓ | Youtube GRAD ↓ | Youtube dtSSD ↓ |
> > | :--- | :--- | ---: | ---: | ---: | ---: | ---: | ---: | ---: | ---: |
> > | TCVOM | Respective Sets | 56.18 | 38.90 | 153.95 | 3.84 | 1.57 | 0.40 | 6.74 | 1.52 |
> > | FTP-VM | Respective Sets | 37.98 | 19.90 | 78.06 | 4.24 | 2.26 | 1.10 | 5.63 | 1.70 |
> > | MAGGIE | Respective Sets | 50.04 | 35.23 | 108.01 | 4.90 | 2.37 | 0.98 | 7.69 | 1.77 |
> > | MatAnyone | Respective Sets | 39.44 | 25.63 | 89.60 | 4.10 | 2.05 | 0.76 | 9.67 | 1.75 |
> > | **Matting Anything 2 (Ours)** | Same as FTP-VM | 18.01 | 4.89 | 26.26 | 2.76 | 1.08 | 0.22 | 2.64 | 1.20 |
> > | **Matting Anything 2 (Ours)** | Respective Sets | 14.72 | 3.70 | 23.54 | 2.65 | 1.16 | 0.24 | 3.07 | 1.20 |
> >
> > We did not adopt the training setting of MatAnyone as requested because the VM800 dataset it utilizes is a private, non-public dataset that is inaccessible to us. We selected FTP-VM for alignment because, as shown in Table 6, its training data composition is the most straightforward, consisting solely of YouTubeVIS, VM108, and D-646.
> >
> >
> > [1] MatAnyone: Stable Video Matting with Consistent Memory Propagation, CVPR 2025
> >
> > [2] MaGGIe: Mask Guided Gradual Human Instance Matting, CVPR 2024
> >
> > [3] End-to-End Video Matting With Trimap Propagation, CVPR 2023
> >
> > [4] One-Trimap Video Matting, ECCV 2022
> >
> > [5] Attention-guided Temporally Coherent Video Object Matting, ACMMM 2021
> >
> > [6] Towards Natural Image Matting in the Wild via Real-Scenario Prior, arxiv

---

> > > ### Comment · Reviewer_Uqg2 · 2025-11-25
> > > **Thanks**
> > >
> > > Hi thanks for the rebuttal. My concerns has been well addressed. I will raise my score.

---

### Official Review · Reviewer_Qe9K · 2025-10-31

**Soundness:** 3
**Presentation:** 4
**Contribution:** 3
**Rating:** 8
**Confidence:** 5

**Summary:**

This paper presents Matting Anything 2, a versatile video matting model designed to overcome the domain-specificity (e.g., human-centric) and restrictive first-frame mask requirements of existing methods. The core technical contributions are twofold. First, a Promptable Dual-mode Decoder (PDD) that jointly predicts segmentation masks and high-quality trimaps, leveraging trimap-based guidance for generalization. Second, a Memory-Separable Siamese (MSS) mechanism that recurrently isolates trimap prediction from interfering mask memory, crucially improving temporal consistency for challenging transparent objects. To validate these contributions, the authors introduce the new, diverse Natural Video Matting (NVM) dataset. Experiments demonstrate that MAM2 significantly outperforms state-of-the-art methods on both diverse natural scenes and human portraits, accepting flexible prompts like points or boxes.

**Strengths:**

1. The paper demonstrates compelling quantitative and qualitative results, significantly outperforming previous state-of-the-art methods.
2. The paper well extends SAM2's promptable, generalist architecture to handle the distinct and more complex task of alpha matting.
3. The paper is well-written, clearly organized, and easy to follow.

**Weaknesses:**

1. The name Natural Video Matting is confusing. In matting literature, "natural" typically implies real-world, non-composited videos. Since NVM is synthetic (composited from assets), this name is a misnomer and should be revised to avoid ambiguity.
2. All experiments are conducted exclusively on synthetic (composited) videos. This leaves a significant gap in evaluation, as performance on real-world videos that contain artifacts like complex lighting, sensor noise, and motion blur remains unproven. The matting "anything" claim is therefore not fully substantiated.
3. The paper lacks a dedicated limitations. There is no discussion of potential failure cases.

**Questions:**

The paper presents an extension of SAM 2 to the matting domain, and the quantitative and qualitative results shown are excellent. My primary question, however, concerns the validation scope. All experiments were conducted on synthetic (composited) videos. This raises a question about the method's true capability to video matting "anything". To fully substantiate the paper's strong claims, I recommend that the authors provide qualitative results (and quantitative results if possible) on real-world video clips, such as those sourced from YouTube, to demonstrate the model's robustness to non-synthetic artifacts.

---

> ### Author Response · Authors · 2025-11-23
> **Official Comment by Authors**
>
> We are very grateful for your positive evaluation of our work, which we find highly encouraging. We have responded to your comments point-by-point and revised the manuscript accordingly. We hope that these revisions satisfactorily address your concerns.
>
> >**W1: the name Natural Video Matting is confusing**
>
> We agree that the original name could lead to confusion regarding the data source. Therefore, we have revised the dataset name to Natural Object Video Matting (NOVM) in the updated manuscript. This nomenclature is intended to emphasize that the dataset focuses on natural object categories, distinguishing the content from human portraits, without suggesting that the videos are non-composited.
>
> >**W2: lack of evaluation on real-world videos**
>
> We strongly agree with the importance of evaluating matting models on real-world videos. However, as discussed in MatAnyone [1], acquiring high-quality annotations for real-world videos is notoriously difficult. Consequently, MatAnyone is also unable to provide quantitative results regarding the model's performance on fine-grained details in real-world videos.
>
> To address this, we have added qualitative results on real-world videos in Figures 11 and 12. All videos in these two figures are real-world videos. Furthermore, we have replaced a subset of the visual examples throughout the manuscript (e.g., Figures 1, 3, and 13) with real-world videos instead of synthetic videos. In addition, for the image matting task, both the test set used for quantitative metrics and the qualitative results in Figure 14 are exclusively based on real-world images.
>
> >**W3: lacks a dedicated limitations**
>
> We agree that including a section on limitations is necessary. Therefore, we have added a new Limitations section in Appendix E. In this section, we discuss the challenges MAM2 faces with highly transparent objects (e.g., dissipating smoke) and the issue of background color bleeding during foreground extraction (e.g., bubbles), identifying the recovery of clean foregrounds from alpha mattes as an open problem for future exploration. More details can be found in  Appendix E.
>
> >**Q1: Performance on real-world videos**
>
> refer to W2
>
>
> [1] MatAnyone: Stable Video Matting with Consistent Memory Propagation, CVPR 2025

---

### Official Review · Reviewer_Fzs3 · 2025-11-03

**Soundness:** 3
**Presentation:** 3
**Contribution:** 3
**Rating:** 6
**Confidence:** 4

**Summary:**

This paper tackles the problem of generalized video matting beyond human-centric domains.
It introduces Matting Anything 2, a robust model capable of handling diverse objects, including transparent ones, with flexible user prompts such as points, boxes, or masks.
The authors propose a Promptable Dual-mode Decoder (PDD) that jointly predicts segmentation masks and trimaps to enhance matting quality and generalization.
To address temporal instability for transparent objects, a Memory-Separable Siamese (MSS) mechanism is designed.
Extensive experiments on diverse exiting benchmark and the newly proposed Natural Video Matting (NVM) dataset demonstrate that MAM2 achieves state-of-the-art accuracy and strong generalization to diverse, real-world scenes.

**Strengths:**

1. The paper is overall easy to follow and understand. It provides a clear motivation and explains key ideas effectively with well-designed supporting figures (e.g., Figure 3 and 4).

2. The ablation studies are comprehensive and convincingly demonstrate the effectiveness of each proposed component.

3. The authors conduct extensive evaluations across diverse tasks (image and video matting) and environments, which strongly support the generality and robustness of the proposed model.

**Weaknesses:**

1. Although the model shows superior performance over baselines across multiple tasks, much of the improvement may stem from leveraging the powerful foundation model SAM2. The proposed method benefits greatly from SAM2’s strong generalization and semantic understanding, whereas most baselines are not based on such advanced foundation models. Therefore, a more in-depth comparison with other SAM1/SAM2-based matting models is essential. For the image matting task, paper [A], which also utilizes SAM, would be a particularly relevant and strong comparison.

2. The Natural Video Matting (NVM) dataset is presented as one of the main contributions, but its description lacks sufficient detail. While brief statistics (in Table 1) and a few visual examples (in the supplementary material) are provided, the paper should offer a more detailed breakdown of the dataset composition—such as domain categories and their relative proportions—especially since it emphasizes dataset diversity as a key feature.

3. Since the primary application of video matting lies in video editing, it would strengthen the paper if the authors demonstrated editing results using the generated alpha mattes, rather than only presenting matte outputs.

[A] ZIM: Zero-Shot Image Matting for Anything, ICCV 2025

**Questions:**

Minor issues include a few typos (e.g., “iamge” in line 406) and missing citations (e.g., MEMatte in line 203 is mentioned without a proper reference). A careful proofread and citation check would improve the paper’s overall quality.

---

> ### Author Response · Authors · 2025-11-23
> **Official Comment by Authors**
>
> We are very grateful for your positive evaluation of our work, which we find highly encouraging. We have responded to your comments point-by-point and revised the manuscript accordingly. We hope that these revisions satisfactorily address your concerns.
>
> >**W1: Comparison with other SAM1/SAM2-based matting models:**
>
> Your intuition is well-founded; the robust generalization and semantic understanding capabilities of SAM 1/2 can indeed significantly enhance matting models. Therefore, to ensure a fair comparison, the methods listed in Table 4 have included SAM-based approaches: specifically, Matting Anything is built upon SAM 1, while SEMatte utilizes SAM 2. To improve clarity, we have explicitly indicated which methods are SAM-based in the image matting experiments section of the revised manuscript.
>
> Following your suggestion, we have also provided the results for ZIM in the table below, as it is also based on SAM 1. However, it is important to note that, unlike all other compared methods, ZIM incorporates the AIM-500 test set into its training process. This inconsistency in training settings was the primary reason for its initial exclusion from our comparative analysis. Nevertheless, we recognize ZIM as a significant SAM-based matting method. We have supplemented the discussion at the beginning of Section 4.4 to include ZIM.
>
> | Method | MSE | SAD | Grad | Conn |
> | :--- | ---: | ---: | ---: | ---: |
> | ZIM | 47.76 | 92.33 | 23.05 | 20.18 |
> | Matting Anything | 11.60 | 36.66 | 21.04 | 18.99 |
> | SmartMatting | 7.65 | 25.33 | 27.16 | 13.54 |
> | SEMatte | 7.65 | 24.30 | 16.06 | 13.64 |
> | SDMatte | 4.91 | 19.81 | 15.84 | 11.97 |
> | **Matting Anything 2 (Ours)** | **4.24** | **18.07** | **13.88** | **11.01** |
>
>
> >**W2: Breakdown of the dataset composition:**
>
> We have supplemented Table 9 with the sample counts and proportions for different categories in the dataset, as shown below. We agree that this breakdown effectively enables readers to quickly grasp the diversity and class balance of our dataset.
>
> | Category | Animals | Bubble | Cloud | Explosion | Fire | Frost | Plant | Slime | Vehicles | Water | Sum |
> | :--- | :---: | :---: | :---: | :---: | :---: | :---: | :---: | :---: | :---: | :---: | :---: |
> | Quantity | 9 | 4 | 4 | 5 | 4 | 4 | 7 | 2 | 4 | 7 | 50 |
> | Proportions | 18% | 8% | 8% | 10% | 8% | 8% | 14% | 4% | 8% | 14% | 100% |
>
>
> >**W3: Demonstration of editing results with our method:**
>
> Following your suggestion, we have presented several matting-based editing effects in Figure 15. Since video matting empowers creators with control at the layer and object levels, it serves as a cornerstone technology for visual effects in the film industry. This includes operations such as background replacement and the manipulation of object trajectories and scales. Furthermore, by independently controlling the speed and chronological order of foreground and background events, it enables sophisticated time manipulation effects, such as slow motion and the time inversion effect famously employed by Christopher Nolan. The examples discussed herein represent only a fraction of the creative possibilities enabled by matting, and experienced and creative practitioners can undoubtedly leverage matting to generate far more sophisticated and visually stunning effects.
>
> >**Q1: Typos and missing citations:**
>
> We sincerely apologize for the errors resulting from our oversight. We have thoroughly proofread the entire manuscript and corrected these mistakes. We appreciate your kind suggestion.

---

### Author Response · Authors · 2025-12-02
**Official Comment by Authors**

We sincerely thank the Area Chair and Reviewers for their valuable time and constructive feedback. We have provided point-by-point responses to the reviewers' comments, addressing issues regarding novelty, presentation, and more. Following the discussion, Reviewers Uqg2 and TLqf confirmed that their concerns were resolved and chose to raise their scores on November 25. We thank the reviewers again for their valuable suggestions and timely responses, which have made our work more solid.

---

### Meta-Review · Area_Chair_aJQp · 2026-01-05

**Summary:**

This paper presents Matting Anything 2, a versatile video matting model leveraging SAM2 for flexible user prompts.

Initial reviews highlighted significant concerns regarding novelty relative to SAM2, evaluation fairness, dataset detail, and validation on real-world videos. The authors provided a comprehensive rebuttal, clarifying the architectural novelty of the trimap-based pipeline, adding a detailed dataset breakdown, providing qualitative real-world results, and conducting a comparative experiment with aligned training data. These responses resolved most concerns, e.g., those raised by Reviewers Uqg2 and TLqf.

Key strengths are the effective technical contributions and strong experimental results. A primary unresolved weakness, noted by Reviewer Qe9K, is the lack of quantitative evaluation on non-synthetic real-world videos, which limits full validation of the "matting anything" claim. This gap in real-world performance evidence remains a notable limitation.

**Reviewer Concerns:**

See above.

**Reviewer Scores:**

While some reviewers may revise their scores positively, others may not, as there is a lack of quantitative evaluation on real-world, non-synthetic videos.

---

### Decision · Program_Chairs · 2026-01-26

Accept (Poster)